# The complex relationship of exposure to new *Plasmodium* infections and incidence of clinical malaria in Papua New Guinea

Natalie E Hofmann[1,2], Stephan Karl[3,4], Rahel Wampfler[1,2], Benson Kiniboro[4], Albina Teliki[4], Jonah Iga[4], Andreea Waltmann[3,5], Inoni Betuela[4], Ingrid Felger[1,2], Leanne J Robinson[3,4,5,6†*], Ivo Mueller[3,5,7,8†*]

[1]Swiss Tropical and Public Health Institute, Basel, Switzerland; [2]University of Basel, Basel, Switzerland; [3]Walter and Eliza Hall Institute of Medical Research, Parkville, Australia; [4]Papua New Guinea Institute of Medical Research, Goroka, Papua New Guinea; [5]University of Melbourne, Melbourne, Australia; [6]Burnet Institute, Melbourne, Australia; [7]ISGlobal, Barcelona Centre for International Health Research, Hospital Clínic-University of Barcelona, Barcelona, Spain; [8]Institut Pasteur, Paris, France

*For correspondence:
robinson@wehi.edu.au (LJR);
ivomueller@fastmail.fm (IM)

†These authors contributed equally to this work

Competing interests: The authors declare that no competing interests exist.

**Abstract** The molecular force of blood-stage infection ($_{mol}$FOB) is a quantitative surrogate metric for malaria transmission at population level and for exposure at individual level. Relationships between $_{mol}$FOB, parasite prevalence and clinical incidence were assessed in a treatment-to-reinfection cohort, where *P.vivax* (*Pv*) hypnozoites were eliminated in half the children by primaquine (PQ). Discounting relapses, children acquired equal numbers of new *P. falciparum* (*Pf*) and *Pv* blood-stage infections/year ($Pf$-$_{mol}$FOB = 0–18, $Pv$-$_{mol}$FOB = 0–23) resulting in comparable spatial and temporal patterns in incidence and prevalence of infections. Including relapses, $Pv$-$_{mol}$FOB increased >3 fold (relative to PQ-treated children) showing greater heterogeneity at individual ($Pv$-$_{mol}$FOB = 0–36) and village levels. *Pf*- and *Pv*-$_{mol}$FOB were strongly associated with clinical episode risk. Yearly *Pf* clinical incidence rate (IR = 0.28) was higher than for *Pv* (IR = 0.12) despite lower *Pf*-$_{mol}$FOB. These relationships between $_{mol}$FOB, clinical incidence and parasite prevalence reveal a comparable decline in *Pf* and *Pv* transmission that is normally hidden by the high burden of *Pv* relapses.
Clinical trial registration: ClinicalTrials.gov NCT02143934
DOI: https://doi.org/10.7554/eLife.23708.001

## Introduction

Renewed emphasis on malaria control has resulted in substantial reductions in overall malaria prevalence and incidence in many endemic countries (*World Health Organization, 2015*). However, where transmission persists, it is highly heterogeneous even on small spatial scales (*Bousema et al., 2012*). Individual exposure is further influenced by factors such as use of bednets, attractiveness to mosquitoes, or behavioural differences. In Papua New Guinea (PNG), malaria prevalence has sharply declined in the last decade, largely as a result of two nationwide distributions of long-lasting insecticide treated bednets (LLIN) (*Hetzel et al., 2015*; *Hetzel et al., 2014*; *Hetzel et al., 2012*). *P. vivax* and *P. falciparum* PCR-prevalence in the general population was reduced from 32% and 39% in 2006 to 13% and 18% in 2010 (*Koepfli et al., 2015*). Already before this decline in malaria prevalence, studies in PNG had reported significant heterogeneity in malaria transmission attributed to local population structure and geographical diversity (*Hetzel et al., 2015*; *Cattani et al., 1986*; *Genton et al., 1995*; *Müller et al., 2003*; *Mueller et al., 2009a*).

**eLife digest** Malaria is caused by five different species of parasites that are transmitted to humans by bites from parasite-carrying mosquitos. Once in human blood, the parasites rapidly multiply. People who live in countries where malaria is common may become infected and never show any symptoms because their immune systems are able to keep parasite numbers low. Repeated infections, or infection with more than one species of malaria parasite also are common. Some species of malaria, including *Plasmodium vivax*, can hibernate in the liver for weeks or months after the infection and only become active later.

Asymptomatic infections, multi-parasite infections, and reactivating parasites make it hard to measure how often new malaria infections occur. One way scientists can determine if a new infection has occurred is by genotyping the parasites in a person's blood. Genotyping involves looking for small differences in the parasite DNA. For example, a study in Papua New Guinea, where *P. vivax* is very common, showed that reactivations of hibernating parasites were more common than new infections.

Now, Hofmann et al. use the same study in Papua New Guinea to compare the frequency and consequences of new infections with *P. vivax* and another malaria parasite, *Plasmodium falciparum*. In the study, 466 children from 6 villages were followed for 8 months with tests every 2 to 4 weeks to genotype the parasites in their blood. Some of the children were treated with antimalarial drugs to help wipe out any existing parasites including hibernating ones. While *P. vivax* was about twice as common in blood samples—likely due to reactivation—genotyping showed that new infections with the two parasites occur at equal rates and often at the same times and locations.

Hofmann et al. also showed that some villages and some children had much higher rates of infection than others. This difference could not fully be explained by use of bednets or other preventive measures. Children were more likely to become ill from *P. falciparum* than *P. vivax* even though *P. vivax* was more common. But children with more frequent infections with *P. falciparum* seemed better able to manage the parasites and were less likely to develop symptoms that those with infrequent infections. The experiments show that genotyping may help scientists better track new malaria infections and develop better strategies to prevent or treat malaria.
DOI: https://doi.org/10.7554/eLife.23708.002

Prior to the up-scaling of malaria control, *P. vivax* endemicity in PNG was among the highest worldwide (*Hetzel et al., 2015*). Clinical immunity to *P. vivax* was acquired very rapidly in PNG children, and the incidence of *P. vivax* clinical episodes peaked in children younger than two years with only very few *P. vivax* clinical episodes reported in children older than 5 years or adults (*Genton et al., 2008*; *Michon et al., 2007*; *Lin et al., 2010*; *Betuela et al., 2012*). In contrast, the risk for uncomplicated *P. falciparum* clinical episodes increased during early childhood (*Lin et al., 2010*) and significant reductions in incidence of clinical episodes or high-density infections were only observed in children aged 5 years and older (*Michon et al., 2007*). Compared to the incidence of clinical malaria, prevalence of *P. falciparum* and *P. vivax* peaked in older age groups, with asymptomatic infections remaining common until adulthood in PNG (*Koepfli et al., 2015*; *Mueller et al., 2009a*). Concordant with the species-specific pattern in the burden of clinical episodes, *P. vivax* prevalence peaked in younger age groups than *P. falciparum* prevalence (*Koepfli et al., 2015*; *Mueller et al., 2009a*).

As malaria transmission declines, it is important to understand the resulting changes in malaria prevalence and clinical incidence patterns, as well as the extent of heterogeneity in transmission within malaria endemic regions so that high-risk areas can be identified and targeted (*Mosha et al., 2014*). Most attempts to delineate high and low transmission areas made to-date, by both researchers and control programs, have used passive case surveillance or cross-sectional malaria indicator surveys. These surveillance strategies result in clinical incidence and prevalence estimates, both of which are surrogate markers for transmission. A more accurate understanding of the relationship between exposure to new infections and malaria prevalence or clinical incidence is needed to determine how accurately these surrogate markers represent heterogeneity in transmission at local scales. In addition, quantifying clinical incidence in relation to exposure to blood-stage

infections can increase our insight into the development and maintenance of immunity to malaria in a setting of sustained malaria control (**Battle et al., 2015**; **Cameron et al., 2015**).

The molecular force of blood-stage infection ($_{mol}$FOB) describes the number of new genotypes observed in consecutive blood samples from cohort participants over time (**Mueller et al., 2012**; **Koepfli et al., 2013**). Genotyping of highly polymorphic markers detects superinfecting parasite clones in asymptomatic (but parasitaemic) or symptomatic individuals. $_{mol}$FOB thus provides a longitudinal, individual and quantitative measure for exposure to new blood-stage malaria infections (**Mueller et al., 2012**; **Koepfli et al., 2013**). For *P. falciparum*, $_{mol}$FOB is closely linked to the number of infective mosquito bites and therefore is a direct proxy for the actual force of infection (FOI) and thus for transmission in endemic settings (**Smith et al., 2010**). For *P. vivax*, clones appearing in the blood-stream can either originate directly from an infective mosquito bite or from a relapsing liver hypnozoite (**Koepfli et al., 2013**). For *P. vivax*, $_{mol}$FOB is thus a compound measure of exposure to newly acquired infections from mosquito bites and relapsing blood-stage infections.

The usefulness of $_{mol}$FOB as a surrogate marker of individual exposure was validated originally in a cohort of young PNG children 1–4 years of age, in which species-specific $_{mol}$FOB was the most important predictor of clinical incidence for both species (**Mueller et al., 2012**; **Koepfli et al., 2013**). Although some spatial heterogeneity of transmission was observed in that study for both species, due to the high level of transmission the species-specific difference in rate of immune acquisition was the predominant feature in that study. While *P. vivax* $_{mol}$FOB (*Pv*-$_{mol}$FOB) did not change with age, the incidence of *P. vivax* clinical episodes decreased significantly with age, with a faster rate of decrease in children with high *Pv*-$_{mol}$FOB [12]. *P. falciparum* $_{mol}$FOB (*Pf*-$_{mol}$FOB) in that cohort was lower compared to *Pv*-$_{mol}$FOB and the incidence of *P. falciparum* clinical episodes increased in parallel with an increasing *Pf*-$_{mol}$FOB in children 1–3 years, reaching a plateau thereafter (**Lin et al., 2010**; **Mueller et al., 2012**). These earlier results indicated that (i) immunity to *P. vivax* is acquired more rapidly in children with higher cumulative exposure, that (ii) this developing immunity led to proportionally fewer clinical *P. vivax* episodes in older children despite similar exposure to new *P. vivax* blood-stage infections, and that (iii) higher exposure to *P. vivax* blood-stage infections, compared to *P. falciparum*, resulted in a more advanced immunity to *P. vivax* in this age group compared to *P. falciparum* (**Doolan et al., 2009**; **Longley et al., 2016**). The major challenge to these cross-species comparisons lies within the intrinsic differences of *Pf*- and *Pv*-$_{mol}$FOB: whereas

**Table 1.** Characteristics of study participants by village.

| Village | N | % female | Mean age (±SD) | Mean weight (±SD) | % LLIN use at enrolment[*] | Mean LLIN use during follow-up[†] (%, range) | Mean Hb (±SD) |
|---------|---|----------|----------------|-------------------|--------------------------|---------------------------------------------|---------------|
| Amahup | 119 | 53 | 7.6 (±1.5) | 19.8 (±3.3) | 99 | 99 (50–100) | 11.1 (±1.0) |
| Albinama | 99 | 43 | 7.7 (±1.5) | 20.0 (±3.3) | 95 | 97 (78–100) | 11.7 (±1.8) |
| Balanga | 54 | 59 | 7.8 (±1.6) | 19.8 (±4.3) | 96 | 99 (83–100) | 11.3 (±1.1) |
| Balif | 93 | 51 | 7.8 (±1.5) | 20.3 (±3.3) | 91 | 99 (69–100) | 11.7 (±1.2) |
| Bolumita | 70 | 50 | 7.4 (±1.7) | 19.3 (±2.9) | 77 | 92 (56–100) | 10.7 (±1.0) |
| Numangu | 31 | 55 | 7.4 (±1.6) | 19.2 (±4.6) | 100 | 100 (92–100) | 12.1 (±1.4) |
| **Total** | **466** | **51** | **7.6 (±1.5)** | **19.8 (±3.5)** | **93** | **100 (50–100)** | **11.4 (±1.4)** |

* LLIN use in the night preceding enrolment.

† Information on LLIN use in the previous night was collected at each follow-up visit and averaged across follow-up per participant. Mean LLIN use by village was calculated from the averaged individual LLIN use.

Hb: Haemoglobin.

DOI: https://doi.org/10.7554/eLife.23708.003

**Table 2.** *Plasmodium* infection status at enrolment by village.

| | P. falciparum | | | | | P. vivax | | | | | P. malariae | | | P. ovale | | |
|---|---|---|---|---|---|---|---|---|---|---|---|---|---|---|---|---|
| Village | N pos. | Prevalence by qPCR (CI95) | % mixed* | Mean† density (IQR) | Mean MOI‡ (range) | N pos. | Prevalence by qPCR (CI95) | % mixed* | Mean† density (IQR) | Mean MOI‡ (range) | N pos. | Prevalence by qPCR (CI95) | % mixed* | N pos. | Prevalence by qPCR (CI95) | % mixed* |
| Albinama | 18 | 18 (11–27) | 72 | 131 (38–189) | 1.4 (1–4) | 54 | 55 (44–65) | 24 | 3 (1–17) | 1.8 (1–7) | 9 | 9 (5–17) | 67 | 5 | 5 (2–12) | 100 |
| Amahup | 14 | 12 (7–19) | 57 | 56 (14–105) | 1.6 (1–5) | 46 | 39 (30–48) | 24 | 3 (1–29) | 2.2 (1–7) | 12 | 10 (6–17) | 83 | 0 | 0 | |
| Balanga | 15 | 28 (17–42) | 67 | 79 (30–848) | 1.7 (1–5) | 23 | 43 (30–57) | 43 | 2 (1–28) | 2.0 (1–7) | 9 | 17 (8–30) | 56 | 2 | 3 (0–14) | 50 |
| Balif | 8 | 9 (4–17) | 63 | 64 (10–325) | 2.0 (1–4) | 35 | 38 (30–48) | 14 | 2 (1–14) | 1.9 (1–6) | 7 | 8 (3–15) | 57 | 0 | 0 | |
| Bolumita | 50 | 71 (59–81) | 80 | 331 (62–1988) | 2.2 (1–8) | 47 | 67 (55–78) | 81 | 3 (2–27) | 2.9 (1–10) | 28 | 40 (29–52) | 89 | 8 | 11 (5–22) | 100 |
| Numangu | 8 | 26 (13–45) | 75 | 192 (30–848) | 1.1 (1–2) | 18 | 58 (39–75) | 28 | 3 (1–25) | 1.6 (1–5) | 4 | 13 (4–31) | 50 | 0 | 0 | |
| Overall | 113 | 24 (20–28) | 73 | 163 (20–1103) | 1.9 (1–8) | 223 | 48 (43–52) | 37 | 3 (1–23) | 2.2 (1–10) | 69 | 15 (12–18) | 75 | 15 | 3 (2–5) | 93 |
| p-value§ | | <0.001 | 0.034 | 0.086 | 0.047 | | <0.001 | <0.001 | 0.947 | 0.020 | | <0.001 | 0.086 | | <0.001 | 0.133 |

\* % of infections by qPCR that are mixed-species infections.

† Geometric mean of species-specific *18S rRNA* copy numbers per µl blood.

‡ MOI, multiplicity of infection: number of *Pf-msp2* and *Pv-msp1F3* alleles per infection.

§ Differences between villages were tested for using Chi² and Fisher's exact test (prevalence, proportion mixed) or Kruskal-Wallis test (MOI, log10-transformed parasite density).

DOI: https://doi.org/10.7554/eLife.23708.004

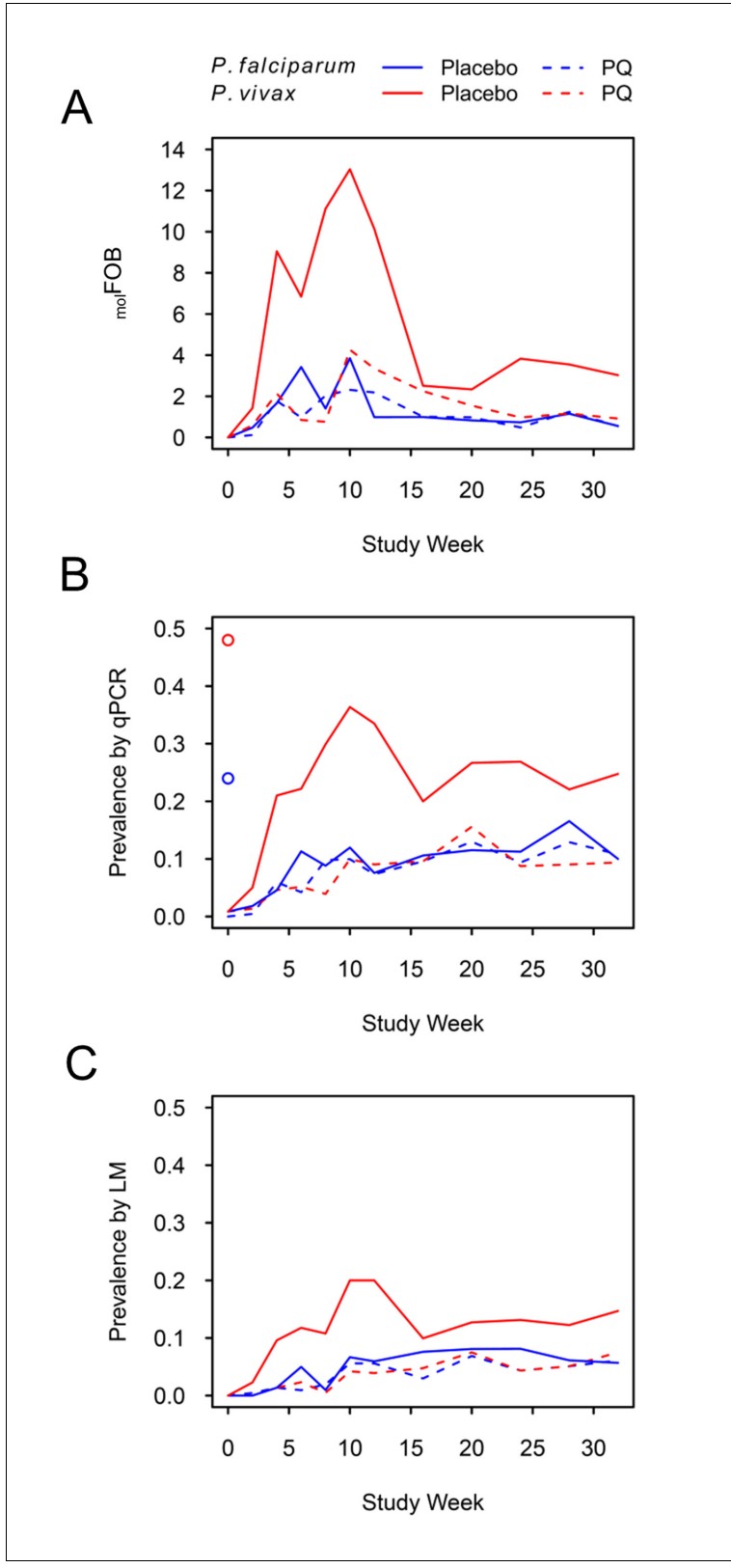

**Figure 1.** *P. falciparum* and *P. vivax* ₘₒₗFOB (A), prevalence by qPCR (B) and LM (C) by week of follow-up. Blue lines, *P. falciparum*; red lines, *P. vivax*; solid lines, placebo arm; dashed lines, PQ arm. Open circles in (B) mark enrolment qPCR prevalence for each species.

DOI: https://doi.org/10.7554/eLife.23708.005

*Figure 1 continued on next page*

*Figure 1 continued*

The following figure supplements are available for figure 1:

**Figure supplement 1.** Definition of new infections for calculating $_{mol}$FOB.

DOI: https://doi.org/10.7554/eLife.23708.006

**Figure supplement 2.** *P. ovale* and *P. malariae* prevalence by qPCR during follow-up.

DOI: https://doi.org/10.7554/eLife.23708.007

*Pf*-$_{mol}$FOB is a direct marker of mosquito-borne transmission, *Pv*-$_{mol}$FOB is a composite measure reflecting both newly acquired infections and those caused by relapses of previously acquired infections.

In this study, we extend the analysis of $_{mol}$FOB's relationship with incidence of clinical malaria episodes to older PNG children and a lower transmission scenario. In addition, given the unique study design that randomized blood-stage only or blood- plus liver-stage treatment at enrolment (*Robinson et al., 2015*), we are now, for the first time, able to compare the incidence of newly acquired *P. falciparum* infections with both the incidence of newly acquired *P. vivax* infections and relapsing *P. vivax* infections. Advancing on previous studies that investigated each species individually (*Mueller et al., 2012*; *Koepfli et al., 2013*), we now provide a comparative analysis of *P. falciparum* and *P. vivax*, directly exploring the role of exposure to multiple *Plasmodium* species in the development of clinical immunity to *P. falciparum* and *P. vivax* malaria. We quantify in detail the extent of heterogeneity in $_{mol}$FOB on a small geographical scale and relate this to heterogeneity in clinical episode incidence to investigate effects of small-scale variation in malaria transmission on local malaria epidemiology. By combining in-depth molecular parasitological data with demographic and clinical data, this study thus provides detailed insights into the changing epidemiology of malaria in PNG in response to intense malaria control efforts.

**Table 3.** Multivariable predictors for time to recurrent blood-stage infection with *Plasmodium* species by qPCR

| Variable | *P. vivax* | | | *P. falciparum* | | | *P. malariae* | | | *P. ovale* | | |
|---|---|---|---|---|---|---|---|---|---|---|---|---|
| | AHR[*] | CI$_{95}$ | *p*-value | AHR[*] | CI$_{95}$ | *p*-value | AHR[*] | CI$_{95}$ | *p*-value | AHR[*] | CI$_{95}$ | *p*-value |
| PQ treatment | 0.18 | 0.13–0.25 | <0.001 | 0.73 | 0.52–1.02 | 0.064 | 0.51 | 0.22–1.19 | 0.121 | 0.31 | 0.12–0.75 | 0.010 |
| Age | 0.95 | 0.87–1.04 | 0.247 | 1.05 | 0.94–1.17 | 0.361 | 0.98 | 0.75–1.29 | 0.905 | 0.96 | 0.74–1.26 | 0.793 |
| LLIN use at enrolment | 0.62 | 0.39–0.98 | 0.043 | 0.84 | 0.49–1.44 | 0.531 | 1.33 | 0.33–6.09 | 0.715 | 0.95 | 0.26–3.43 | 0.936 |
| Hb at enrolment (g/dl) | 0.88 | 0.80–0.98 | 0.019 | 0.90 | 0.80–1.02 | 0.099 | 0.83 | 0.61–1.12 | 0.224 | 0.92 | 0.66–1.28 | 0.634 |
| *Village* | | | | | | | | | | | | |
| Albinama (ref) | 1 | | | 1 | | | 1 | | | 1 | | |
| Amahup | 0.45 | 0.29–0.71 | 0.001 | 0.58 | 0.31–1.11 | 0.101 | 0.34 | 0.07–1.79 | 0.205 | 2.83 | 0.29–27.48 | 0.370 |
| Balanga | 2.15 | 1.40–3.31 | <0.001 | 1.81 | 0.99–3.30 | 0.054 | 0.92 | 0.24–3.60 | 0.910 | 7.74 | 0.85–70.45 | 0.070 |
| Balif | 1.00 | 0.66–1.54 | 0.983 | 0.60 | 0.30–1.19 | 0.145 | 0.24 | 0.03–2.07 | 0.193 | 4.60 | 0.51–41.41 | 0.173 |
| Bolumita | 3.34 | 2.09–5.33 | <0.001 | 4.73 | 2.69–8.30 | <0.001 | 1.21 | 0.34–4.31 | 0.770 | 19.43 | 2.19–172.37 | 0.008 |
| Numangu | 0.83 | 0.44–1.59 | 0.583 | 2.29 | 1.17–4.50 | 0.015 | 0.82 | 0.15–4.53 | 0.823 | 3.17 | 0.19–52.41 | 0.420 |
| *Infection status at enrolment (by qPCR)* | | | | | | | | | | | | |
| Uninfected (ref) | 1 | | | 1 | | | 1 | | | 1 | | |
| *P. vivax* | 1.27 | 0.91–1.78 | 0.165 | 1.37 | 0.86–2.20 | 0.186 | 0.92 | 0.20–4.18 | 0.913 | 2.17 | 0.68–6.97 | 0.192 |
| *P. falciparum* | 1.36 | 0.84–2.19 | 0.205 | 1.56 | 0.86–2.82 | 0.145 | 3.54 | 0.85–14.72 | 0.083 | 1.25 | 0.26–5.90 | 0.779 |
| *P. malariae* | 0.83 | 0.38–1.85 | 0.655 | 0.99 | 0.38–2.56 | 0.977 | 6.35 | 1.31–30.81 | 0.022 | 1.58 | 0.17–14.30 | 0.676 |
| Mixed *P.f.* or *P.v.*[†] | 1.74 | 1.14–2.65 | 0.010 | 2.08 | 1.25–3.48 | 0.005 | 3.37 | 0.88–12.90 | 0.076 | 2.03 | 0.55–7.53 | 0.287 |

[*] AHRs were modeled using Cox proportional hazard regression.

[†] Mixed infection including *P. falciparum* or *P. vivax* infection in conjunction with one or more other *Plasmodium spp.*

PQ: Primaquine; LLIN: long-lasting insecticide-treated net; Hb: haemoglobin.

DOI: https://doi.org/10.7554/eLife.23708.008

# Results

## Demographic and parasitological parameters at enrolment

This study was conducted in six villages in Maprik district, East Sepik Province, PNG between August 2009 and May 2010 (*Robinson et al., 2015*). 524 children aged 5–10 years were enrolled and randomized to receive either chloroquine (CQ), artemeter-lumefantrine (AL) and primaquine (PQ); or CQ, AL and placebo. Demographic parameters of the 466 children that completed the full course of randomized treatment with PQ/CQ/AL (n = 233) or placebo/CQ/AL (n = 233), and were thereafter closely followed for 8 months, were comparable between the six villages (*Table 1*).

*P. vivax* was the most common infection at enrolment with 48% of children positive by quantitative PCR (qPCR), followed by *P. falciparum* (24%), *P. malariae* (15%) and *P. ovale* (3%; *Table 2*). 39% of children were not infected with any *Plasmodium* species at enrolment. The vast majority of *P. malariae* (75%) and almost all *P. ovale* infections (93%) occurred in children co-infected with either *P. vivax* and/or *P. falciparum* (*Table 2*). Prevalence of each *Plasmodium* species varied between villages (*P. falciparum*, 9–71%; *P. vivax*, 38–67%; *P. malariae*, 8–40%; *P. ovale*, 0–11%; *Table 2*) and was highest in Bolumita for all species. Accordingly, mixed-species infections were also most prevalent in Bolumita (*Table 2*). The multiplicity of infection (MOI), that is, the number of parasite genotypes per infection, also varied between villages for both species (mean *P. falciparum* MOI, 1.1–2.2 clones/infection; mean *P. vivax* MOI, 1.6–2.9 clones/infection) and children from Bolumita carried more multi-clone infections with *P. vivax* and *P. falciparum* than children in other villages (*Table 2*). Mean *P. falciparum* parasite density was almost two- to six-fold higher in Bolumita (331 18S rRNA gene copies/µl) than in other villages (56–192 18S rRNA gene copies/µl, *Table 2*).

## $_{mol}$FOB and parasite prevalence after randomized radical cure treatment

Children who had received PQ for clearance of *P. vivax* hypnozoites experienced similar numbers of new blood-stage infections with *P. falciparum* and *P. vivax* during follow-up (mean *Pf*-$_{mol}$FOB = 1.5

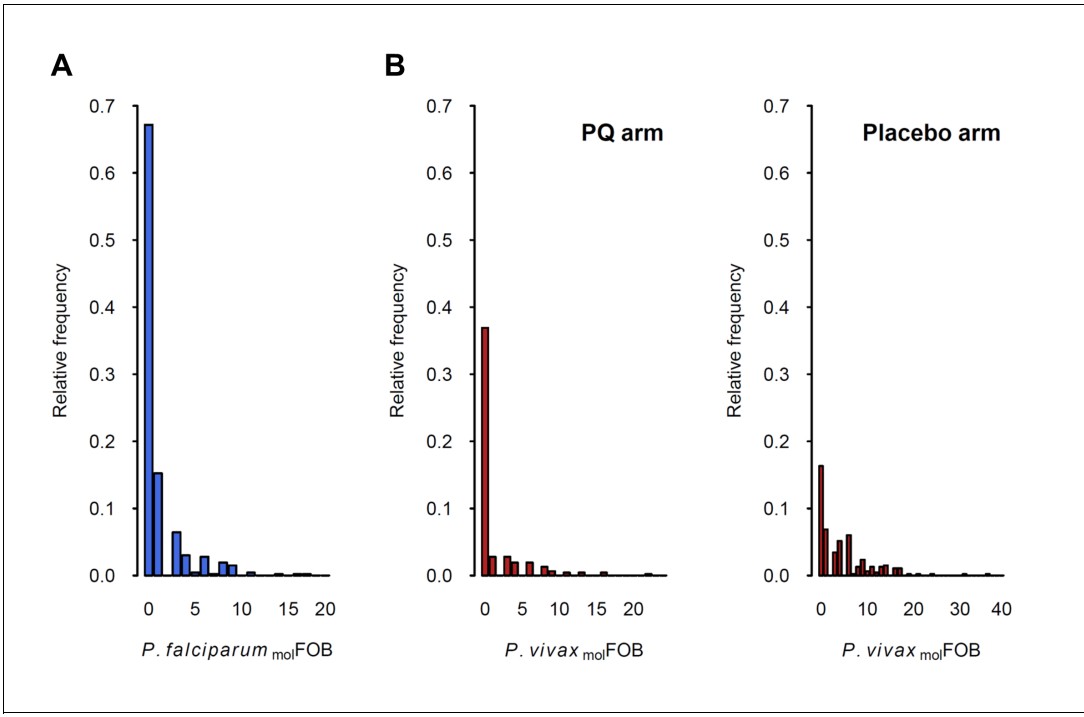

**Figure 2.** Distribution of *P.falciparum* $_{mol}$FOB (**A**) and *P. vivax* $_{mol}$FOB by treatment arm (**B**). Relative frequencies among the 466 children are shown.
DOI: https://doi.org/10.7554/eLife.23708.009

CI$_{95}$ [1.3–1.7] new blood-stage clones/year, $Pv$-$_{mol}$FOB = 1.6 [1.4–1.9] new blood-stage clones/year, *Figure 1A*, *Figure 1—figure supplement 1*). $Pf$-$_{mol}$FOB in the placebo arm was comparable to the PQ arm (mean $Pf$-$_{mol}$FOB = 1.4 [1.2–1.6] new blood-stage clones/year), whereas due to the hypnozoite reservoir $Pv$-$_{mol}$FOB was more than three times higher in the placebo arm compared to the PQ arm (mean $Pv$-$_{mol}$FOB = 5.4 [4.9–5.8] new blood-stage clones/year). $Pv$-$_{mol}$FOB in the placebo arm showed a pronounced peak at months 2–3 of follow-up, which likely represents a wave of fast-relapsing hypnozoites in children who did not receive PQ (*Figure 1A*).

*P. vivax* prevalence in the PQ arm was comparable to *P. falciparum* prevalence throughout the study and increased steadily, irrespective of the diagnostic method used (*Figure 1B and C*). *P. vivax* prevalence increased more rapidly in the placebo arm until month 3 of follow-up and dropped thereafter, similar to patterns in $Pv$-$_{mol}$FOB in the same arm. Prevalence as measured by qPCR did not reach pre-treatment levels until the end of the study for any of the four *Plasmodium* species (*Figure 1B*, *Figure 1—figure supplement 2*).

At the end of follow-up, *P. vivax* prevalence by qPCR in the placebo arm was 25% [19–31%], and therefore more than two-fold higher than in the PQ arm (9% [6–14%]; *Figure 1B*). Also, throughout follow-up, *P. vivax* prevalence in the placebo arm was 2–3 fold higher compared to the PQ arm, suggesting that at least 50% of the overall *P. vivax* prevalence in this cohort can be attributed to the contribution of relapses.

Similarly, throughout and at the end of follow-up *P. vivax* prevalence in the placebo arm was 2–3 fold higher compared to *P. falciparum* (irrespective of treatment arm; *P. falciparum* prevalence at

**Table 4.** Multivariable predictors of $Pv$- and $Pf$-$_{mol}$FOB per follow-up interval.
Model predictions from this model were used for mapping $_{mol}$FOB in *Figure 3A*.

| | P. vivax | | | | | | P. falciparum | | |
| | PQ arm | | | Placebo arm | | | Combined arms | | |
| Variable | IRR* | CI$_{95}$ | *p*-value | IRR* | CI$_{95}$ | *p*-value | IRR* | CI$_{95}$ | *p*-value |
|---|---|---|---|---|---|---|---|---|---|
| PQ treatment | n.a.† | n.a. | n.a. | n.a. | n.a. | n.a. | 0.89 | 0.65–1.22 | 0.474 |
| New *P. falc.* infections in interval‡ | 1.32 | 0.92–1.89 | 0.134 | 1.10 | 0.85–1.42 | 0.466 | n.a. | n.a. | n.a. |
| New *P. vivax* infections in interval‡ | n.a. | n.a. | n.a. | n.a. | n.a. | n.a. | 1.15 | 0.97–1.36 | 0.100 |
| Age | 0.86 | 0.74–1.01 | 0.059 | 0.95 | 0.87–1.04 | 0.305 | 1.03 | 0.92–1.14 | 0.640 |
| LLIN use at enrolment | 0.96 | 0.51–1.79 | 0.897 | 0.62 | 0.43–0.91 | 0.013 | 1.07 | 0.7–1.62 | 0.755 |
| Hb at enrolment (g/dl) | 0.85 | 0.72–1.01 | 0.063 | 0.91 | 0.85–0.99 | 0.025 | 0.85 | 0.75–0.97 | 0.013 |
| *Village* | | | | | | | | | |
| Albinama (ref) | 1 | | | 1 | | | 1 | | |
| Amahup | 0.02 | 0–0.11 | <0.001 | 0.56 | 0.34–0.91 | 0.020 | 0.52 | 0.25–1.07 | 0.074 |
| Balif | 0.85 | 0.4–1.8 | 0.664 | 1.74 | 1.16–2.61 | 0.007 | 1.81 | 0.98–3.35 | 0.059 |
| Balanga | 0.28 | 0.1–0.82 | 0.020 | 1.13 | 0.73–1.73 | 0.590 | 0.75 | 0.37–1.52 | 0.423 |
| Bolumita | 1.52 | 0.73–3.17 | 0.268 | 2.67 | 1.83–3.9 | <0.001 | 6.05 | 3.32–11.05 | <0.001 |
| Numangu | 0.5 | 0.15–1.68 | 0.264 | 0.76 | 0.4–1.43 | 0.394 | 2.8 | 1.39–5.64 | 0.004 |
| *Study Day* | | | | | | | | | |
| Day 0–35 (ref) | 1 | | | 1 | | | 1 | | |
| Day 36–80 | 1.37 | 0.54–3.48 | 0.509 | 1.99 | 1.39–2.84 | <0.001 | 2.42 | 1.44–4.07 | 0.001 |
| Day 81–175 | 1.34 | 0.57–3.12 | 0.503 | 0.89 | 0.61–1.3 | 0.538 | 1.13 | 0.7–1.84 | 0.616 |
| Day > 175 | 0.65 | 0.25–1.69 | 0.374 | 0.56 | 0.38–0.83 | 0.004 | 0.87 | 0.48–1.56 | 0.643 |

*IRRs were modeled per sampling interval using negative binomial generalized estimating equations allowing for repeated visits with log-link and an exchangeable correlation structure.

† n.a., not applicable.

‡$_{mol}$FOB in the follow-up interval (time-varying covariate).

PQ: Primaquine; LLIN: long-lasting insecticide-treated net; Hb: haemoglobin.

DOI: https://doi.org/10.7554/eLife.23708.011

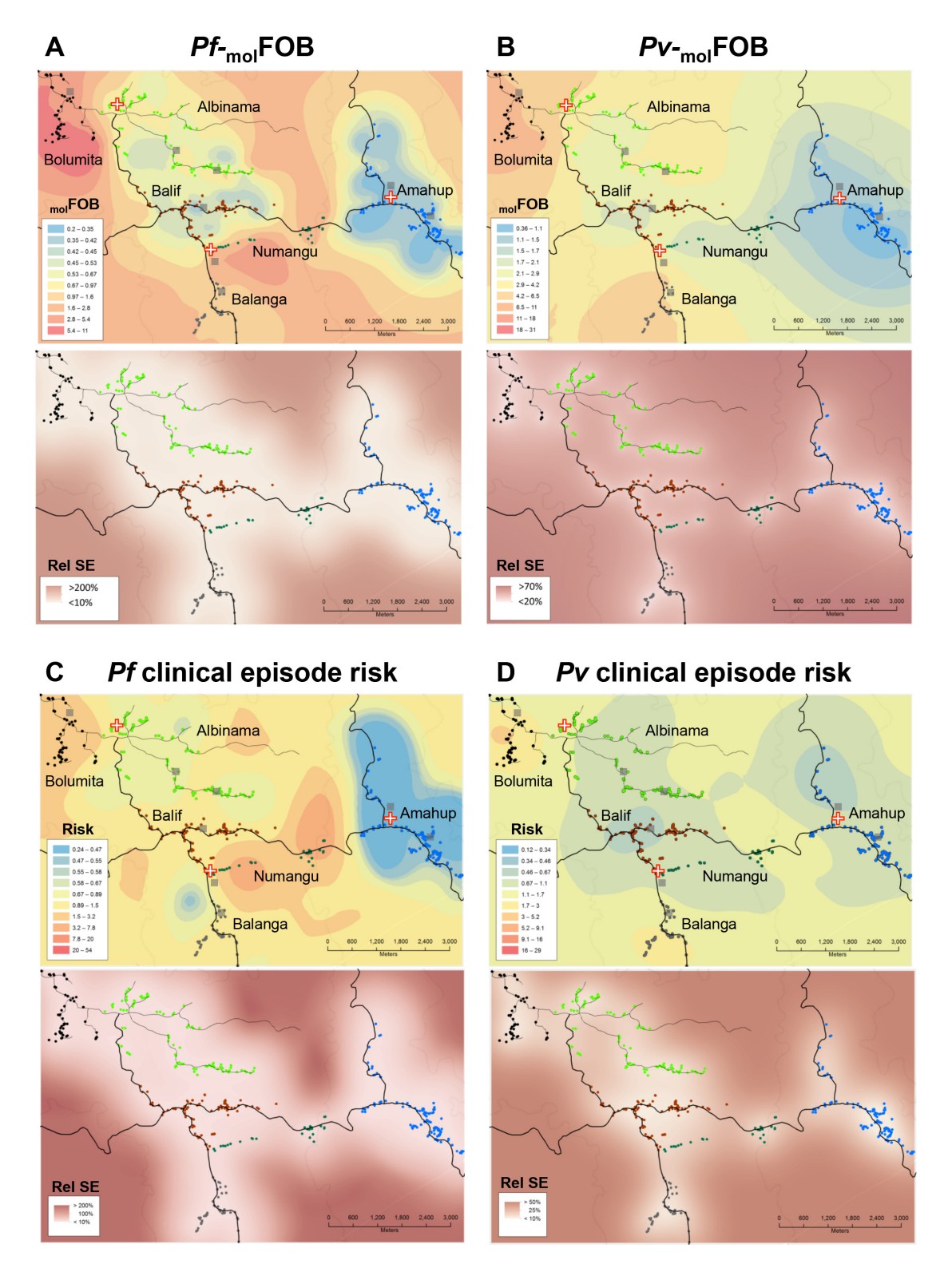

**Figure 3.** Heterogeneity in molFOB (A, B) and clinical episode risk (C, D) of *P.falciparum* (A, C) and *P. vivax* (B, D). Upper panels show the kriging fit of model predictions of molFOB and clinical episode risk of children in both treatment arms. Lower panels show the standard error relative to the kriging estimate. Dots represent study participants' houses and are color-coded according to village. Black lines: vehicle-accessible road; dark grey lines:

*Figure 3 continued on next page*

*Figure 3 continued*
vehicle-inaccessible road; light grey lines: river; red/white cross: health center or aid post; grey square: school or enrolment location. Maps were prepared using ArcGIS 10.2 (Esri, USA).
DOI: https://doi.org/10.7554/eLife.23708.010

end of follow-up, 10% [8–14%]), which is in agreement with the prevalence pattern at enrolment. Assuming equal transmission from mosquitoes for both species, which was corroborated by a comparable *Pf*-$_{mol}$FOB and *Pv*-$_{mol}$FOB in the PQ arm, *P. vivax* relapses have contributed to a *P. vivax* prevalence twice as high as that of *P. falciparum*.

## Risk of recurrent blood-stage infections and $_{mol}$FOB during follow-up

In the present study design, recurrent blood-stage infection can either originate from a new transmission event (both arms and all species) or for *P. vivax* and *P. ovale* also from a relapse of any previous infection (placebo arm only). After adjusting for the effect of PQ treatment (*Robinson et al., 2015*), village of residence and infection status by qPCR at enrolment were the main predictors for the risk of recurrent *Plasmodium spp.* during follow-up (*Table 3*). Interestingly, in addition to a protective effect against recurrent *P. vivax* and *P. ovale*, the risk of recurrent *P. falciparum* was also reduced by 27% [0–48%] after PQ treatment (p=0.064).

The risk of a recurrent infection (measured by qPCR) with *P. falciparum*, *P. vivax* and *P. ovale* varied more than 7-fold between villages, with a higher risk observed in Bolumita (78%, 77%, and 15% with recurrent *P. vivax*, *P. falciparum* and *P. ovale*, respectively) compared to the other villages (recurrent *P. vivax*, range 25–73%; recurrent *P. falciparum*, range 12–44%; recurrent *P. ovale*, range 0–7%). For *P. falciparum* and *P. vivax*, a mixed infection at enrolment as measured by qPCR was further associated with up to a two-fold increased risk of recurrent infection (*P. falciparum*: AHR = 2.08 [1.25–3.48], p=0.005; *P. vivax*: AHR = 1.74 [1.14–2.65], p=0.010; *Table 3*), supporting the idea that focal transmission within villages leads to the presence of high-risk and low-risk individuals. For *P. malariae*, the infection status at enrolment was a stronger predictor of risk of recurrent infection than village of residence. An infection with *P. falciparum*, *P. malariae* or a mixed infection at enrolment as measured by qPCR was associated with up to a 6-fold increase in risk of recurrent *P. malariae* (AHR$_{Pf\text{-enrol}}$ = 3.54 [0.85–14.72], p=0.083; AHR$_{Pm\text{-enrol}}$ = 6.35 [1.31–30.81], p=0.022; AHR$_{mixed}$ = 3.37 [0.88–12.90], p=0.076; *Table 3*).

Reported use of a LLIN during the night previous to enrolment was associated with a reduced risk of recurrent *P. vivax* and *P. falciparum* in univariate analyses (*Supplementary file 1* - Table 1) but to a lesser extent in multivariable analyses (*P. vivax*: AHR = 0.62 [0.39–0.98], p=0.043, *P. falciparum*: AHR = 0.84 [0.49–144], p=0.531). Haemoglobin (Hb) level at enrolment was negatively associated with the risk of recurrent infection with *P. vivax* (AHR = 0.88 [0.80–0.98], p=0.019) and *P. falciparum* (AHR = 0.90 [0.80–1.02], p=0.099). Patterns in the risk of recurrent infections with *P. falciparum* and *P. vivax* as measured by light microscopy (LM, *Supplementary file 2*) were similar to those observed for re-infection as measured by qPCR. When based on LM observation (but not as measured by qPCR), increasing age was associated with a reduced risk of recurrent *P. vivax* (AHR = 0.85 [0.77–0.95], p=0.004; *Supplementary file 2*) but an increased risk of recurrent *P. falciparum* (AHR = 1.16 [1.01–1.33], p=0.037; *Supplementary file 2*).

The incidence of new *P. falciparum* and *P. vivax* blood-stage clones detected during follow-up, that is $_{mol}$FOB, was highly variable between individual children and ranged from 0 to 18 new clones/year for *P. falciparum* (*Figure 2A*) and 0 to 36 or 23 new blood-stage clones/year for *P. vivax* in the placebo or PQ arm, respectively (*Figure 2B*). Mean *Pf*- and *Pv*-$_{mol}$FOB varied significantly between villages and were higher in Bolumita (*Pf*-$_{mol}$FOB = 4.9 new blood-stage clones/year, *Pv*-$_{mol}$FOB$_{PQ}$ $_{arm}$=4.4 new blood-stage clones/year, *Pv*-$_{mol}$FOB$_{placebo}$ $_{arm}$=12.1 new blood-stage clones/year; *Table 4*; *Figure 3*) than in the other villages (*Pf*-$_{mol}$FOB, range 0.7–1.8 new blood-stage clones/year; *Pv*-$_{mol}$FOB$_{PQ}$ $_{arm}$, range 0.03–2.2 new blood-stage clones/year; *Pv*-$_{mol}$FOB$_{placebo}$ $_{arm}$, range 2.3–7.4 new blood-stage clones/year). In univariate analyses, new *P. vivax* infections were strongly associated with new *P. falciparum* infections per sampling interval and vice versa, suggesting concurrent exposure to the two species (*Supplementary file 1* – Table 2). However, these effects were reduced when other variables of varying exposure such as village of residence or infection at enrolment were

**Table 5.** Multivariable predictors for time to *P. vivax* and *P. falciparum* clinical episodes.
Model predictions from this model were used for mapping the relative risk of clinical malaria episodes in *Figure 3C and D*.

| Variable | *P. vivax* | | | *P. falciparum* | | |
|---|---|---|---|---|---|---|
| | AHR* | CI$_{95}$ | *p*-value | AHR* | CI$_{95}$ | *p*-value |
| PQ treatment | 0.76 | 0.34–1.68 | 0.497 | 1.79 | 1.05–3.03 | 0.031 |
| *P. vivax* $_{mol}$FOB‡ | 1.07 | 1.04–1.09 | <0.001 | n.a. | n.a. | n.a. |
| *P. falciparum* $_{mol}$FOB‡ | n.a. | n.a. | n.a. | 1.15 | 1.11–1.21 | <0.001 |
| Age | 0.62 | 0.46–0.84 | 0.002 | 0.98 | 0.85–1.13 | 0. 799 |
| LLIN use at enrolment | 0.84 | 0.24–2.88 | 0.778 | 0.44 | 0.22–0.87 | 0.018 |
| Hb at enrolment (g/dl) | 0.95 | 0.74–0.67 | 0.668 | 0.85 | 0.71–1.01 | 0.070 |
| *Village* | | | | | | |
| Albinama (ref) | 1 | | | 1 | | |
| Amahup | 0.89 | 0.23–3.46 | 0.871 | 0.65 | 0.20–2.08 | 0.465 |
| Balif | 1.48 | 0.45–4.86 | 0.518 | 1.26 | 0.50–3.14 | 0.626 |
| Balanga | 0.85 | 0.21–3.53 | 0.827 | 1.39 | 0.59–3.30 | 0.455 |
| Bolumita | 0.99 | 0.24–4.03 | 0.987 | 1.32 | 0.58–3.03 | 0.508 |
| Numangu | 1.00 | 0.23–4.31 | 0.997 | 4.29 | 2.06–8.97 | <0.001 |
| *Infection status at enrolment (by qPCR)* | | | | | | |
| Uninfected (ref) | 1 | | | 1 | | |
| *P. vivax* | 0.77 | 0.29–2.07 | 0.608 | 1.64 | 0.91–2.95 | 0.101 |
| *P. falciparum* | 1.74 | 0.59–5.11 | 0.316 | 0.97 | 0.34–2.77 | 0.954 |
| Mixed *P.f. or P.v.* | 1.59 | 0.56–4.50 | 0.381 | 1.24 | 0.57–2.68 | 0.582 |

* AHRs were modeled using multiple failure Cox proportional hazard regression.

† n.a., not applicable

‡ Average $_{mol}$FOB until the time of failure (time-varying covariate).

PQ: Primaquine; LLIN: long-lasting insecticide-treated net; Hb: haemoglobin.

DOI: https://doi.org/10.7554/eLife.23708.013

included in the multivariable model (*P. vivax*: IRR$_{PQ\ arm}$=1.32 [0.92–1.89], p=0.134; IRR$_{Placebo\ arm}$=1.10 [0.85–1.42], p=0.466; *P. falciparum*: IRR = 1.15 [0.97–1.36], p=0.100, *Table 4*). LLIN use,

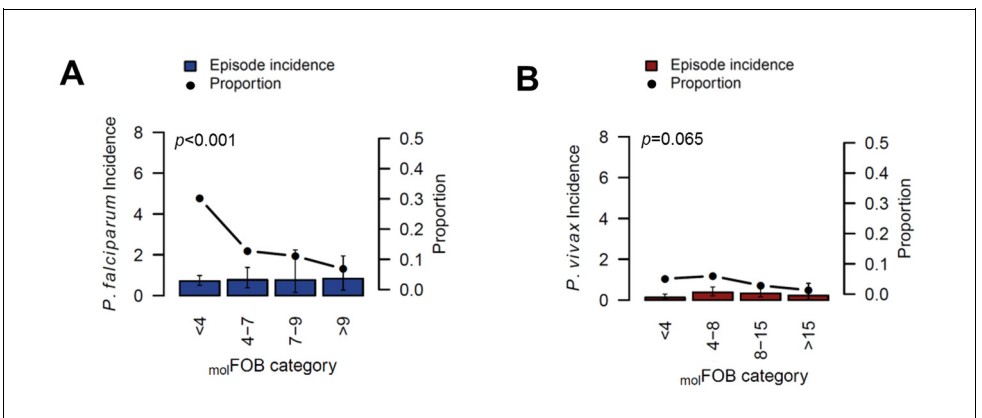

**Figure 4.** The incidence of *P.falciparum* (A) and *P. vivax* (B) clinical episodes relative to $_{mol}$FOB. Mean clinical episode incidence is shown as bars (left axis) and proportion of clinical episode incidence divided by $_{mol}$FOB as connected dots (right axis). Error bars represent 95% CIs. *p*-values refer to the differences between groups in the proportion of clinical episodes and new infections, assessed by Chi$^2$ or Fisher's exact test.

DOI: https://doi.org/10.7554/eLife.23708.014

**Table 6.** Multivariable predictors for odds of *P. falciparum* clinical episodes

| Variable | OR[*] | CI$_{95}$ | *p*-value |
|---|---|---|---|
| | *P. falciparum* episode | | |
| PQ treatment | 1.42 | 0.80–2.52 | 0.226 |
| *P. vivax* qPCR positive[†] | 0.35 | 0.15–0.78 | 0.011 |
| *P. falciparum* $_{mol}$FOB[‡] | 1.21 | 1.10–1.34 | <0.001 |
| Age | 0.93 | 0.80–1.09 | 0.370 |
| LLIN at enrolment | 0.37 | 0.16–0.83 | 0.016 |
| Hb (g/dl) at enrolment | 0.88 | 0.70–1.11 | 0.292 |
| *Village* | | | |
| Albinama (ref) | 1 | | |
| Amahup | 0.41 | 0.12–1.39 | 0.154 |
| Balif | 0.9 | 0.26–3.08 | 0.870 |
| Balanga | 1.19 | 0.42–3.39 | 0.747 |
| Bolumita | 1.48 | 0.42–5.17 | 0.540 |
| Numangu | 4.17 | 1.64–10.58 | 0.003 |
| *Study Day* | | | |
| Day 0–80 | 1 | | |
| Day 81–175 | 0.99 | 0.51–1.91 | 0.972 |
| Day > 175 | 0.83 | 0.39–1.75 | 0.629 |

\* ORs were modeled using a binomial generalized estimating equation with logit link function using an exchangeable correlation structure.

† Determined as *P. vivax* positive at the same or previous sampling visit.

‡ $_{mol}$FOB in the follow-up interval (time-varying covariate).

PQ: Primaquine; LLIN: long-lasting insecticide-treated net; Hb: haemoglobin.

DOI: https://doi.org/10.7554/eLife.23708.015

although strongly associated with lower *Pf*- and *Pv*-$_{mol}$FOB in univariate analyses (***Supplementary file 1*** – Table 2), remained significantly associated in multivariable models only for *P. vivax* in the placebo arm, where sleeping under a LLIN in the night previous to enrolment was associated with a 38% [9–57%] reduction in *Pv*-$_{mol}$FOB (p=0.013, ***Table 4***). Each additional year of age was associated with a 14% [0–26%] reduction *Pv*-$_{mol}$FOB per sampling interval in the PQ arm (p=0.059), while no age effect was observed in the placebo arm or for *P. falciparum* (***Table 4***). Hb level at enrolment was negatively associated with *Pf*- and *Pv*-$_{mol}$FOB (*P. vivax*: IRR$_{PQ\ arm}$=0.85 [0.72–1.01], p=0.063; IRR$_{Placebo\ arm}$=0.91 [0.85–0.99], p=0.025; *P. falciparum*: IRR = 0.85 [0.75–0.97], p=0.013), suggesting anaemia in individuals continuously exposed to blood-stage infections.

## Patterns in the risk of *P. vivax* and *P. falciparum* clinical episodes

A total of 98 clinical malaria episodes, here defined as fever plus presence of LM-detectable parasites, were observed during the study period. Of these, 64 (65%) exceeded the previously established pyrogenic thresholds of 2500 and 500 parasites/µl per LM for *P. falciparum* and *P. vivax*, respectively (***Mueller et al., 2009b***). *P. falciparum* was the most common cause of clinical malaria episodes (*P. falciparum*, 64 clinical episodes; *P. vivax*, 31 clinical episodes; mixed *P. falciparum/P. vivax* by LM, 3 clinical episodes), despite lower incidence of new *P. falciparum* blood-stage clones compared with *P. vivax* (*P. falciparum*, 342 new *P. falciparum* blood-stage clones; *P. vivax*, 849 new blood-stage clones). Including clinical episodes with mixed infection as determined by LM in the estimates for both species, clinical incidence rate (IR) was 0.28 [0.21–0.35] *P. falciparum* episodes/year and 0.12 [0.08–0.17] *P. vivax* episodes/year. At least one new blood-stage clone was detected in 70% (47/67) of samples from *P. falciparum* and 71% (24/34) of samples from *P. vivax* clinical episodes. Of these clinical episodes with new blood-stage clones, 96% (45/47) and 83% (20/24) carried only the new but no persistent *P. falciparum* and *P. vivax* clones, respectively.

*P. vivax* clinical episodes occurred mainly in the placebo arm shortly after directly observed treatment (DOT) (*Robinson et al., 2015*), the time of peak *Pv*-$_{mol}$FOB due to relapsing hypnozoites (*Figure 1A*). On an individual level, *Pv*-$_{mol}$FOB was positively associated with the risk of clinical episodes and each additional blood-stage *P. vivax* clone increased the risk of experiencing a *P. vivax* clinical episode slightly (AHR = 1.07 [1.04–1.09], p<0.001; *Table 5*). No significant differences in *P. vivax* clinical episode risk were observed between villages after adjusting for individual $_{mol}$FOB. The risk for a *P. vivax* clinical episode decreased significantly with age (AHR = 0.62 [0.46–0.84], p=0.002; *Table 5*). This was paralleled by a decrease in *P. vivax* densities with age (by qPCR, exp(β)=0.90 [0.83–0.98], p=0.016; *Supplementary file 3*) indicative of more advanced immunity against *P. vivax* and thus better control of *P. vivax* densities in older children.

Patterns in the occurrence of *P. falciparum* clinical episodes during follow-up were more complex. Between-village variation in *P. falciparum* clinical episode risk remained significant even after adjusting for individual exposure. This effect was mainly apparent in Numangu, where children were at three- to six-fold higher risk for clinical episodes than children in other villages (Numangu AHR = 4.29 [2.06–8.97], other villages range AHR = 0.65 [0.20–2.08] to 1.39 [0.59–3.30]; *Table 5*). Overall, *Pv*-$_{mol}$FOB was positively associated with the risk of clinical episodes and each additional *P. falciparum* blood-stage clone slightly increased the risk for *P. falciparum* clinical episodes (AHR = 1.15 [1.11–1.21], p<0.001; *Table 5*); however, relative to the number of new blood-stage clones, *P. falciparum* clinical episodes were less frequent in highly exposed children compared to low-exposed children (*Figure 4*). One clinical episode per three new blood-stage clones was detected in the least exposed children (*Pf*-$_{mol}$FOB <4 new blood-stage clones/year), but only one clinical episode per 15 blood-stage clones in the highest exposed children (*Pf*-$_{mol}$FOB >9 new blood-stage clones/year, Fisher's exact test p<0.001). Age was not associated with the risk of *P. falciparum* clinical episodes. LLIN use at enrolment was associated with a 56% [13–78%] reduced risk of *P. falciparum* clinical episodes (p=0.018; *Table 5*).

A higher risk for *P. falciparum* clinical episodes in children that had received PQ treatment for clearance of *P. vivax* liver stages was observed (AHR = 1.79 [1.05–3.03], p=0.031; *Table 5*), suggesting a potential protective effect of *P. vivax* infections against *P. falciparum* clinical episodes. Analysis to further explore this revealed that a concurrent or recent infection (i.e., at the same or preceding follow-up visit) with *P. vivax* reduced the odds of a *P. falciparum* clinical episode by 65% [22-85%] (p=0.011; *Table 6*). Further indications for a potential interaction between the two species was also observed when analyzing *P. falciparum* parasite densities, which were reduced by 55% [19–75%] (p=0.008; *Supplementary file 3*) in mixed *P. falciparum/P. vivax* infections compared to *P. falciparum* single infections, indicative of suppression of one of the species in mixed infections.

## Discussion

In the present study, we describe striking heterogeneity in malaria transmission not only between closely neighboring communities in Maprik district, PNG, but also substantial differences in exposure between individual children from the same village. On village level this heterogeneity is apparent both when using traditional markers such as prevalence of infection, as well as when using the novel 'reference standard' marker of individual exposure $_{mol}$FOB. The increased resolution provided by $_{mol}$FOB further allows quantifying heterogeneity in exposure to new blood-stage infections between individual children. Extending an earlier study in a neighboring area in which younger children had been enrolled, and that had identified $_{mol}$FOB as the most important predictor of malaria clinical episodes (*Mueller et al., 2012*; *Koepfli et al., 2013*), we confirmed that $_{mol}$FOB remains significantly associated with the risk for clinical episodes, but other factors such as age (*P. vivax*), or a mixed *Pf/Pv* infection and village factors not captured by any of the other parameters assessed (*P. falciparum*) have a stronger effect on the risk for clinical malaria (*Table 5 and 6*).

Malaria transmission is often estimated by investigating the more accessible human host rather than the mosquito vector (*Tusting et al., 2014*). Because *P. falciparum* blood-stage infections are a direct outcome of mosquito-to-human transmission, infection parameters assessed in the human blood closely reflect *P. falciparum* transmission. In contrast, relapses arising from dormant hypnozoites contribute substantially to *P. vivax* blood-stage infections (*Robinson et al., 2015*), thus complicating the assessment of mosquito-to-human *P. vivax* transmission via infection parameters measured in the human blood. The unique design of this study, that combined clearance of

hypnozoites in half of the study participants with subsequent measurement of $Pv\text{-}_{mol}FOB$, allowed us to identify the burden of *P. vivax* infections due to mosquito-to-human transmission (in hypnozoite-cleared individuals) and compare it to the total burden of *P. vivax* infections. We found a highly similar incidence and comparable temporal and spatial heterogeneity of *P. falciparum* and *P. vivax* infection acquired through renewed exposure to infected mosquito bites. In children that experienced the full burden of relapses we found two-fold higher *P. vivax* infection prevalence and 4-times higher incidence ($_{mol}FOB$) compared to *P. falciparum*. This first quantitative comparative assessment of *P. falciparum* and *P. vivax* transmission using non-entomological molecular parameters thus indicates that the observed differences in epidemiology between the two species are largely due to the high burden of relapsing *P. vivax* blood-stage infections (*Robinson et al., 2015*). Our molecular results thus support recent entomological data from Dreikikir district, 50 km from Maprik in East Sepik Province (*Reimer et al., 2016*) as well as earlier studies in East Sepik (*Hii et al., 2001*) that found similar sporozoite rates for *P. falciparum* and *P. vivax*.

Our previous analysis of this cohort had investigated the contribution of the hypnozoite reservoir to *P. vivax* infection and disease in order to inform strategies for achieving a sustained reduction of the *P. vivax* burden in PNG (*Robinson et al., 2015*). Here, we now describe the post-treatment re-infection dynamics in higher temporal detail. Through a detailed comparison of these patterns for *P. vivax* and *P. falciparum* in PQ and placebo-treated children we further elucidate the contribution of relapses to *P. vivax* prevalence and clinical incidence, which are the most commonly used parameters for planning and monitoring of malaria control strategies. *P. vivax* relapses accounted for more than half of the observed *P. vivax* prevalence in this cohort, which is lower than what was previously estimated as the contribution of relapses towards $Pv\text{-}_{mol}FOB$ by comparison of treatment arms (77%, *Robinson et al., 2015*). This difference can be accounted for by the higher number of *P. vivax* multiple clone infections that will accumulate more rapidly in the placebo-arm, where additional parasite clones from relapses and/or new infections may overlap with without a corresponding change in overall prevalence.

While we previously described a sustained effect of PQ treatment with significant reductions in $Pv\text{-}_{mol}FOB$ observed up to eight months post treatment (*Robinson et al., 2015*), here, we describe temporal variation in relapse rate with a rapid and wave-like recurrence of *P. vivax* in children from the placebo arm, who had retained their hypnozoites (*Figure 1A*). The concurrent, modest peaks in $Pf\text{-}_{mol}FOB$ and $Pv\text{-}_{mol}FOB$ in the PQ arm represent seasonal variation in transmission, which is highest in December and January in the study area (*Mueller et al., 2012*); corresponding to weeks 8–14 of follow-up. A much higher peak and subsequent drop in appearance of new clones within three months after blood-stage only treatment, which was mirrored by a corresponding peak and drop in *P. vivax* prevalence (*Figure 1B andC*), suggests that the incidence of relapse infections in the blood was not constant during follow-up. *P. vivax* infections are often observed following treatment of *P. falciparum* malaria (*Douglas et al., 2011*), and it can thus been hypothesized that the frequency of relapses may be temporarily increased after blood-stage antimalarial treatment (*White and Imwong, 2012*). It is thus conceivable that the blood-stage antimalarial at baseline either triggered *P. vivax* relapses directly, or indirectly by allowing more hypnozoites to establish blood-stage infections in parasite-free hosts. Alternatively, blood-stage *P. vivax* infections from hypnozoites relapsing shortly after baseline treatment (during a period when antimalarial drugs were present at sub-curative levels) may be suppressed to sub-detectable densities until complete waning of drug levels, resulting in simultaneous proliferation and detection of many new blood-stage clones within the first weeks after treatment (*Douglas et al., 2011*; *Tarning et al., 2014*). More detailed modeling of the dynamics of individual *P. vivax* blood-stage infections and their association with potential triggers such as treatment or febrile illness will be required to determine the existence and importance of proposed relapse-triggers.

Malaria transmission showed high micro-spatial heterogeneity with more than 10-fold differences in $Pf\text{-}$ and $Pv\text{-}_{mol}FOB$ (in the PQ arm) between villages despite an overall high LLIN use by the study participants (during follow-up; village average use,>90%; individual use,>50%). Individual LLIN use at enrolment was nevertheless associated with a reduced risk of recurrent *P. falciparum* and *P. vivax* in univariate analyses. However, after adjustment for other related variables (i.e., village of residence or infection status) this association became non-significant. Children living in Bolumita, where both *P. falciparum* and *P. vivax* $_{mol}FOB$ and prevalence were highest, had a modestly lower LLIN use (mean during follow-up, 92%; at enrolment, 77%) compared to children from other villages (mean

during follow-up, 97–100%; at enrolment, 91–100%). It is conceivable that LLIN use in the Bolumita community may be less effective in reducing malaria transmission (*Killeen et al., 2007*; *Smith et al., 2009*). Potential differences between villages in mosquito density, behavior, sporozoite rate, proximity of house or play areas to mosquito breeding sites, or human behavioral factors (related to LLIN use or other risk factors) are however likely to be more important determinants for exposure to infective bites. Small-scale variations in vector species and distribution between and within villages in PNG have been described previously (*Cattani et al., 1986*; *Reimer et al., 2016*; *Charlwood et al., 1986*; *Hii et al., 1997*; *Burkot et al., 1988*) and likely account in a large part for the micro-geographic heterogeneity in malariological parameters observed in this and other studies.

Assessing the incidence of new infections from consecutive blood samples using molecular methods (as is necessary to determine $_{mol}$FOB), is complicated by fluctuating densities of clonal parasitemia that may temporarily fall below the limit of detection of the genotyping PCR, leading to imperfect detectability of clones (*Bretscher et al., 2010*; *Felger et al., 2012*; *Koepfli et al., 2011*). For *P. falciparum*, periodical sequestration of clones and absence from the peripheral blood at time of sampling may further contribute to imperfect detectability. For *P. vivax*, generally low parasite densities aggravate the problem of imperfect detectability, and dis- and re-appearance of clones may be a result of imperfect detectability or relapsing hypnozoites. The overall estimates of $_{mol}$FOB presented here may thus be biased. Accurately assessing the effects of this imperfect detectability on parameters estimated from longitudinal genotyping data, such as $_{mol}$FOB, requires complex mathematical modeling (*Bretscher et al., 2010*; *Felger et al., 2012*; *Sama et al., 2005*; *Sama et al., 2006*). However, although clonal detectability has been shown to decrease with age (*Felger et al., 2012*; *Sama et al., 2006*) and MOI (*Koepfli et al., 2011*), it is unlikely to vary substantially within our cohort's age range and transmission setting. Hence the observed differences in $_{mol}$FOB are likely to accurately reflect the relative differences in individual exposure as well as in population transmission levels within the study area.

Evaluating the impact of malaria control efforts requires monitoring changes in malariological metrics over extended periods of time. Drawing comparisons between studies performed at different times in different age groups is particularly challenging because of the interplay of past and current exposure to infective bites and the resulting anti-malarial immunity in the study population of a certain age. In our cohort, fewer *P. vivax* clinical episodes than *P. falciparum* clinical episodes were detected despite a higher incidence of *P. vivax* blood-stage infections, which is consistent with earlier studies in children of similar age (*Michon et al., 2007*). The very low incidence of clinical *P. vivax* episodes in our cohort, at 0.16 clinical episodes/year (placebo arm [*Robinson et al., 2015*]), contrasts drastically with that of 2.46 *P. vivax* clinical episodes/year observed in an earlier observational cohort of younger children aged 1–4 years from the same area (*Lin et al., 2010*). The 3-fold difference in $Pv$-$_{mol}$FOB between the two cohorts seems modest when compared to the 15-fold difference in the incidence of clinical *Pv* episodes (*Koepfli et al., 2013*). This suggests that the much lower incidence of *P. vivax* clinical illness in 5–10 years old children of this study is more likely explained by an advanced state of immunity to *P. vivax* compared to the younger children of the earlier cohort than the drop in *P. vivax* transmission. Consistently, age emerged as the strongest factor associated with protection against *P. vivax* clinical episodes, $Pv$-$_{mol}$FOB and *P. vivax* parasite density. Like in the previous cohort of younger children from neighboring villages (*Lin et al., 2010*) the incidence clinical *P. vivax* clinical episodes dropped significantly with age. Unlike in the previous cohort of younger children (*Koepfli et al., 2013*), in this cohort we additionally observed a drop in $Pv$-$_{mol}$FOB as well as *P. vivax* densities with age (*Table 4* and *Supplementary file 3*). This is a further indication of the substantial clinical immunity to *P. vivax* acquired during years of past exposure in the children of this study, which is still ongoing after the age of five.

In sharp contrast to the age-dependent decline of *P. vivax* clinical episode incidence, no age-dependent decrease in the incidence of clinical episodes was observed for *P. falciparum*. In a previous cohort study conducted in 2004 in 5–14 year old children in an area from PNG with substantially higher transmission levels (mean incidence risk 5.0 versus 0.8 infections/year, *Michon et al., 2007*; *Robinson et al., 2015*), the risk of moderate- to high-density *P. falciparum* infections decreased significantly with age (*Michon et al., 2007*). Clinical immunity to *P. falciparum* in children of this earlier cohort was not only significantly further advanced compared to children of this cohort, but in addition, transmission was more homogeneous in the area of that study. As a consequence age was a much better marker of life-time exposure and thus immune status compared to the present cohort.

In this cohort, exposure to *P. falciparum* infections was highly heterogeneous between study participants. Mathematical modeling suggests that at heterogeneous transmission, changes of parasite prevalence and clinical episode incidence with age are less pronounced compared to settings with homogeneous transmission (*Ross and Smith, 2010*). Although no age trends were observed for *P. falciparum* in this cohort, when children were stratified into groups ranging from low to high exposure we found that the proportion of *P. falciparum* clinical episodes relative to new infections decreased with increasing exposure. This could either reflect the development of clinical immunity in highly exposed children, or premunition, a proposed mechanism by which established infections help to control superinfections by immunological cross-protection (*Sergent and Parrot, 1935*; *Smith et al., 1999*). In settings of decreasing and heterogeneous transmission, age alone may therefore not be a suitable marker of immunity to *P. falciparum*. Instead, combining age and $_{mol}$FOB to estimate cumulative life-time exposure may provide a more accurate surrogate measure of the extent of acquired clinical immunity. With *P. falciparum* transmission declining in PNG due to successful malaria control strategies (*Koepfli et al., 2015*), it is conceivable that immunity against *P. falciparum* will develop more slowly, shifting the burden of disease towards older age groups or towards more complex, non-linear age patterns. This delay in immune acquisition is however more than compensated by the overall much lower incidence of clinical malaria clinical episodes: in cohort studies in children younger than 4 years from Maprik district, clinical *P. falciparum* incidence had dropped from 2.56 clinical episodes/year before (observational cohort, [*Lin et al., 2010*; *Mueller et al., 2012*]) to 0.67 clinical episodes/year immediately after the free LLIN distribution campaign (placebo arm, [*Betuela et al., 2012*]).

Finally, given that four *Plasmodium* species co-exist in PNG, there has long been considerable interest in potential mechanisms of cross-species immunity and mixed species interactions (*Mueller et al., 2009a*; *Bruce et al., 2000*; *Smith et al., 2001*; *Mehlotra et al., 2000*). However, there is so far no consistent evidence for the presence or absence of cross-protection among *Plasmodium* species. *P. vivax* and *P. falciparum* infections in our study were concentrated in the same children and villages (*Figure 3*), likely due to overlapping focal transmission for *P. falciparum* and *P. vivax* and thus potentially high co-infection rates in mosquitoes. Contrary to an earlier cohort in younger PNG children that found a decreased risk of *P. falciparum* clinical episodes after PQ radical cure (*Betuela et al., 2012*), we found indications for an increased risk of *P. falciparum* illness after clearance of *P. vivax* hypnozoites using PQ. Our data suggests that in individuals with substantial clinical immunity against *P. vivax*, a concurrent *P. vivax* infection may provide protection against *P. falciparum* clinical episodes by limiting *P. falciparum* densities (*Table 6*, *Supplementary file 3*). However, the comparably small number of clinical episodes in this and the earlier contrasting study does not allow an in-depth analysis of causal relationships and therefore does not allow firm conclusions on the potential effects and mechanisms of cross-species interactions in mixed infections.

In conclusion, this study provides detailed insight into the changing epidemiology of malaria in PNG children under sustained malaria control, by using $_{mol}$FOB as a powerful measure to quantitatively investigate patterns of new mosquito-derived *P. falciparum* and *P. vivax* infections versus those for *P. vivax* relapsing infections, as well as spatial and age trends in exposure to these infections. Striking heterogeneity in malaria transmission between villages as well as in individual exposure to new *P. falciparum* and *P. vivax* infections persisted in our study area despite very high use of LLINs. This presents a significant challenge for on-going malaria control efforts. The comparable patterns of new mosquito-derived *P. falciparum* and *P. vivax* infections indicate that sustained use of LLINs does result in a comparable reduction in transmission of both species. The higher incidence and prevalence of *P. vivax* infections observed in our data is thus directly linked to its ability to cause relapsing infections, highlighting the crucial role of hypnozoites for *P. vivax* epidemiology and the need to effectively intervene against these hidden stages. Together, these insights provide a crucial link to evaluate the level of *P. vivax* mosquito-based transmission against that of *P. falciparum* and serve to calibrate other standard malaria indicators such as parasite prevalence or incidence of clinical episodes and to ultimately inform new approaches to surveillance and response systems.

## Materials and methods

### Study design and participants

This study was conducted in six villages in the Albinama and Balif areas, Maprik district, East Sepik Province, PNG between August 2009 and May 2010. The area is serviced by the Albinama health sub-center, Balif aid post and a network of health workers in all study villages. The study design has been described in detail elsewhere (*Robinson et al., 2015*). Briefly, 524 children aged 5–10 years whose parents provided written informed consent for their participation were enrolled and randomized to receive either chloroquine (CQ, days 1–3, total dose 25 mg/kg), artemeter-lumefantrine (Coartem, AL, days 11–13, 2 mg/kg A, 12 mg/kg L) and primaquine (PQ, days 1–20, 0.5 mg/kg/day); or CQ (days 1–3), AL (days 11–13), and placebo (days 1–20) over 20 days of directly observed treatment ($DOT_{1-20}$) in a double-blinded manner. Children were actively visited and examined for signs and symptoms of malaria fortnightly at their schools for 8 months. In addition, passive surveillance was provided by the local health centre, aid post and village health workers throughout the study period. Finger-prick blood samples (250 µl) were collected at fortnightly active-follow-up visits in the first 12 weeks and monthly thereafter, as well as from symptomatic children detected during active or passive morbidity surveillance. Symptomatic children were tested for malaria infection with rapid diagnostic test (RDT, CareStartMalaria pLDH/HRP2 Combo, AccessBio, USA), and only RDT and or LM-confirmed *Plasmodium* infections of any density were treated with a 3 day course of AL.

Household, village and health facility location data was collected using a handheld GPS receiver (Garmin GPSmap62sc) and maps were prepared using ArcGIS 10.2 (Esri, USA).

The study received ethical clearance from the PNG IMR Institutional Review Board (0908), the PNG Medical Advisory Committee (09.11), the Ethics Committee of Basel 237/11 and was conducted in full concordance with the Declaration of Helsinki. The study was registered on ClinicalTrials.gov (NCT02143934).

### Laboratory methods

All blood samples were examined by LM and qPCR for detection and speciation of *Plasmodium* infections as described earlier (*Robinson et al., 2015*). Each blood slide was read independently by two skilled microscopists and re-read by an expert microscopist in case of discrepancies in positivity, speciation or density ($\geq2$ x $\log_{10}$ difference). Thick blood films were examined by LM for 200 fields (1000x magnification) before being declared parasite-negative. Parasite density was converted from the number of parasites per 200–500 white blood cells (WBC) to parasites/µl assuming 8000 WBC/µl (WHO malaria microscopy training guide) and calculated as the geometric mean of all positive reads.

DNA was extracted from the red blood cell pellet using the FavorPrep 96-well genomic DNA extraction kit (Favorgen). Samples carrying any *Plasmodium spp.* infection were identified using a generic qPCR (*Wampfler et al., 2013*) and positives were subsequently tested in species-specific qPCRs (*Rosanas-Urgell et al., 2010*; *Wampfler et al., 2013*). All qPCRs targeted the *small subunit (18S) ribosomal RNA* gene and were performed as simplex (*P. vivax* and *P. falciparum*) or duplex qPCR (*P. malariae*, *P. ovale*). The concentration of target copies per µl of DNA was determined relative to a dilution row of standard plasmid as previously described (*Rosanas-Urgell et al., 2010*). The qPCR limit of detection (LOD) was determined using a standard plasmid dilution row and defined as the last point with more than 50% of replicates positive. The LOD was 2 target copies/µl DNA, equaling 4 target copies/reaction, for all qPCRs. All samples that crossed the fluorescence threshold were scored as positive for species-specific qPCRs. In all samples positive in *P. falciparum* and/or *P. vivax* qPCRs, individual parasite clones were distinguished by genotyping the length-polymorphic *Pf-msp2* or *Pv-msp1F3* marker genes using capillary electrophoresis for highly precise fragment sizing (*Koepfli et al., 2013*; *Koepfli et al., 2011*; *Falk et al., 2006*; *Schoepflin et al., 2009*). MOI was determined by counting the number of detected *Pf-msp2* or *Pv-msp1F3* alleles per sample. $_{mol}$FOB was calculated from the number of new parasite clones detected per child or per sampling interval in the peripheral blood, divided by the individual time at risk or length of the interval. A new infection was defined as a *Pf-msp2* or *Pv-msp1F3* allele not present in the two preceding genotyping-positive samples collected during active or passive surveillance (*Figure 1—figure supplement 1*). Imperfect diagnostic detectability was not further adjusted for.

## Statistical analysis

Children were considered at risk for clinical malaria clinical episodes until the end of the study or until they were censored (on the last visit before two consecutively missed scheduled follow-up visits [*Robinson et al., 2015*]). For clinical endpoints, time-at-risk (TAR) was not further adjusted for interim missed follow-up visits because the intense active and passive case detection presumably led to detection of all malaria clinical episodes. In contrast, TAR for analysis of molecular data (e.g., $_{mol}$FOB) was reduced by the duration of the missed interval if a child was not seen by the study team for six weeks or more ($\geq$42 days). Children with a TAR of less than 3 months (<84 days) were excluded. This resulted in an analyzed population of 466 children (characterized in *Table 1*) of which 430 (92.3%) completed the whole follow-up period, with a median of 15 (IQR: 13–17) study contacts and mean TAR of 186 days (IQR 168–223 days).

Time to first *Plasmodium* infection by qPCR and LM and its association with covariates were modeled using Cox regression, and the proportional hazards assumption was checked using the test based on the Schoenfeld residuals. Multiple failure Cox regression was used to model the time to *P. vivax* and P. *falciparum* clinical episodes.

For statistical analysis, a malaria clinical episode was defined as fever (>37.5°C axillary) plus the presence of LM-detectable parasites, irrespective of RDT result or antimalarial treatment during the field visit. Negative binomial generalized estimating equations (GEE) with log link function using an exchangeable correlation structure were used to model incidence of new infections with *P. falciparum* and *P. vivax* per sampling interval. For these analyses, the time at risk was restricted to the intervals where $_{mol}$FOB could be estimated (i.e., starting in the third follow-up interval). A binomial GEE with logit link function using an exchangeable correlation structure was used to model the odds of a *P. falciparum* clinical episode per interval. Gaussian GEEs with log link function using an exchangeable correlation matrix were used to model log-transformed qPCR parasite densities in qPCR-positive samples, measured as 18S rRNA copy numbers/µl blood. In the GEE and Cox models where $_{mol}$FOB was a covariate, it was included as a time-varying covariate. When modelling $_{mol}$FOB (*Table 4*) and the odds of clinical episodes (*Table 6*) using GEEs, $_{mol}$FOB was calculated for each follow-up interval and used as predictor. In Cox models investigating the risk of clinical episodes (*Table 5*), $_{mol}$FOB was calculated based on the new infections up to the time of failure and used as predictor. In exploratory preliminary analyses we tested for a wide variety of interactions between covariates including interactions between all combinations of $_{mol}$FOB, enrolment infection status, age, village and bednet-usage. All analyses were done using STATA v14 and R.

Maps were drawn using Arcgis 10.1 (Esri Inc.). Ordinary kriging was used to generate the contour maps. Semivariograms were used as the mathematical forms used to express autocorrelation. Input variables for the spatial models were (i)$_{mol}$FOB (prediction of independent variable, $_{mol}$FOB) using the model presented in *Table 4* (negative binomial GEE) for *Pf* and *Supplementary file 4* for *Pv* (same as that shown in *Table 4* but with primaquine and placebo arms combined), resulting in *Figure 3* Panels A and B; (ii) relative risk of clinical episodes as predicted by the model shown in *Table 5*, resulting in Panels C and D of *Figure 3*. Relative standard error maps were generated by dividing the absolute standard error map by the model prediction map.

## Acknowledgements

We sincerely thank the children, their parents and guardians, school principals, teachers, and communities for their willingness to be involved in this study. We are grateful to the staff at Albinama Health Centre and village-based health workers for their assistance. We also wish to thank the field team, administration and laboratory staff at Maprik branch, as well as the molecular parasitology laboratory staff at Goroka branch of PNG IMR for their efforts in sample collection and processing. We thank Matthew Phillip for assistance with GPS data collection. We thank Amanda Ross for statistical advice, as well as Jessica Brewster and Cristian Koepfli for assistance with qPCR. Funding was obtained from the Swiss National Science Foundation (grant no. 310030–134889 and 310030–159580), the International Centers of Excellence in Malaria Research (grant U19 AI089686), and the TransEpi consortium funded by the Bill and Melinda Gates, the NHMRC (#1021544) and the Cellex Foundation. This work was also made possible through Victorian State Government Operational Infrastructure Support and Australian Government NHMRC IRIISS. LJR was supported by an NHMRC Early Career Fellowship #1016443. SK is supported by an NHMRC Early Career Fellowship

#1052760. IM is supported by an NHMRC Senior Research Fellowship (#1043345). The funders had no role in study design, data collection and analysis, decision to publish, or preparation of the manuscript.

## Additional information

### Funding

| Funder | Grant reference number | Author |
|---|---|---|
| National Health and Medical Research Council | Early Career Fellowship #1052760 | Stephan Karl |
| National Institute of Allergy and Infectious Diseases | South West Pacific International Centers of Excellence in malaria research U19 AI089686 | Inoni Betuela<br>Ingrid Felger<br>Leanne J Robinson<br>Ivo Mueller |
| Bill and Melinda Gates Foundation | TransEpi consortium | Inoni Betuela<br>Ingrid Felger<br>Leanne J Robinson<br>Ivo Mueller |
| National Health and Medical Research Council | Project Grant #1021544 | Inoni Betuela<br>Ingrid Felger<br>Leanne J Robinson<br>Ivo Mueller |
| Fundación Cellex | | Inoni Betuela<br>Ivo Mueller |
| Schweizerischer Nationalfonds zur Förderung der Wissenschaftlichen Forschung | 310030-134889 310030-159580 | Ingrid Felger<br>Ivo Mueller |
| National Health and Medical Research Council | Early Career Fellowship #1016443 | Leanne J Robinson |
| National Health and Medical Research Council | Senior Research Fellowship #1043345 | Ivo Mueller |

The funders had no role in study design, data collection and interpretation, or the decision to submit the work for publication.

### Author contributions

Natalie E Hofmann, Data curation, Formal analysis, Investigation, Methodology, Writing—original draft; Stephan Karl, Formal analysis, Writing—original draft; Rahel Wampfler, Investigation, Methodology, Writing—review and editing; Benson Kiniboro, Albina Teliki, Data curation, Project administration, Writing—review and editing; Jonah Iga, Andreea Waltmann, Data curation, Methodology, Writing—review and editing; Inoni Betuela, Supervision, Investigation, Project administration, Writing—review and editing; Ingrid Felger, Conceptualization, Supervision, Investigation, Methodology, Writing—review and editing; Leanne J Robinson, Conceptualization, Data curation, Formal analysis, Supervision, Investigation, Writing—original draft, Project administration, Writing—review and editing; Ivo Mueller, Conceptualization, Formal analysis, Supervision, Funding acquisition, Investigation, Writing—review and editing

### Author ORCIDs

Natalie E Hofmann http://orcid.org/0000-0001-6282-5364
Leanne J Robinson http://orcid.org/0000-0001-9903-1023

### Ethics

Clinical trial registration: ClinicalTrials.gov NCT02143934
Human subjects: The study received ethical clearance from the PNG IMR Institutional Review Board (0908), the PNG Medical Advisory Committee (09.11), the Ethics Committee of Basel 237/11 and

was conducted in full concordance with the Declaration of Helsinki. Written informed consent was obtained from the parents/guardians of all children enrolled in the study.

## Decision letter and Author response

Decision letter https://doi.org/10.7554/eLife.23708.024
Author response https://doi.org/10.7554/eLife.23708.025

# Additional files

## Supplementary files

• Supplementary file 1. Univariate factors. Table 1. Univariate/PQ-adjusted predictors for time to recurrent blood-stage infection with *Plasmodium* species by qPCR. Table 2. Univariate (*P. vivax*) and univariate/PQ-treatment-adjusted (*P. falciparum*) predictors for *Pv*- and *Pf*-$_{mol}$FOB by follow-up interval. Table 3. Univariate/PQ-treatment-adjusted predictors for time to *P. vivax* and *P. falciparum* episodes. Table 4. Univariate/PQ-treatment-adjusted predictors for odds of *P. falciparum* clinical episodes.
DOI: https://doi.org/10.7554/eLife.23708.012

• Supplementary file 2. Multivariable predictors for time to recurrent blood-stage infection with *Plasmodium* species by LM.
DOI: https://doi.org/10.7554/eLife.23708.016

• Supplementary file 3. Multivariable predictors for *P. falciparum* and *P. vivax* density by qPCR during follow-up.
DOI: https://doi.org/10.7554/eLife.23708.017

• Supplementary file 4. Multivariable predictors of *Pv*-$_{mol}$FOB (combining primaquine and placebo arms) per follow-up interval. This model is similar to that presented in *Table 4* in the main text but combines the treatment arms for *P. vivax*. Model predictions from this model were used for mapping molFOI in *Figure 3B*.
DOI: https://doi.org/10.7554/eLife.23708.018

• Transparent reporting form
DOI: https://doi.org/10.7554/eLife.23708.019

## Major datasets

The following dataset was generated:

| Author(s) | Year | Dataset title | Dataset URL | Database, license, and accessibility information |
|---|---|---|---|---|
| Natalie E Hofmann, Stephan Karl, Rahel Wampfler, Benson Kiniboro, Albina Teliki, Jonah Iga, Andreea Waltmann, Inoni Betuela, Ingrid Felger, Leanne J Robinson, Ivo Mueller | 2017 | Data from: The complex relationship of exposure to new Plasmodium infections and incidence of clinical malaria in Papua New Guinea | http://dx.doi.org/10.5061/dryad.f9154 | Available at Dryad Digital Repository under a CC0 Public Domain Dedication |

The following previously published dataset was used:

| Author(s) | Year | Dataset title | Dataset URL | Database, license, and accessibility information |
|---|---|---|---|---|
| Robinson LJ, Wampfler R, Betuela I, Karl S, White MT, Li Wai Suen CSN, Hofmann NE, Kiniboro B, Waltmann A, Brewster J, Lorry L, Tarongka N, Samol L, Silkey M, Bassat Q, Siba PM, Schofield L, Felger I, Mueller I | 2015 | Data from: Strategies for understanding and reducing the Plasmodium vivax and Plasmodium ovale hypnozoite reservoir in Papua New Guinean children: a randomised placebo-controlled trial and mathematical model | http://dx.doi.org/10.5061/dryad.m1n03 | Available at Dryad Digital Repository under a CC0 Public Domain Dedication |

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
