## [Decision Letter]

Thank you for submitting your article "The complex relationship of exposure to new *Plasmodium* infections and incidence of clinical malaria in Papua New Guinea" for consideration by *eLife*. Your article has been reviewed by three peer reviewers, and the evaluation has been overseen by a Reviewing Editor and Prabhat Jha as the Senior Editor. The reviewers have opted to remain anonymous.

The reviewers have discussed the reviews with one another and the Reviewing Editor has drafted this decision to help you prepare a revised submission.

Summary:

This paper presents results of a cohort study in Papua New Guinea where 524 children from six villages were randomised to either chloroquine, artemeter-lumefantrine and placebo (which aims to clear Plasmodium parasites from the blood) or to chloroquine, artemeter-lumefantrine and primaquine (which aims to clear Plasmodium parasites from the blood and *P. vivax* hypnozoites from the liver). Individuals were followed-up for eight months for signs and symptoms of malaria. Repeated blood samples were taken and used to determine the molecular force of blood-stage infection, a proxy measure for malaria transmission and exposure. In the primaquine arm (blood- plus liver-stage treatment), the force of molecular bloodstream infection for *P. vivax* and *P. falciparum* was similar, but both varied substantially between villages and individuals (ranging from 0-23 and 0-18 new clones/year). A higher force of molecular bloodstream infection was seen for *P. vivax* in the blood-stage only arm (0-36 new clones/yr), presumably as a result of relapsing hypnozoites in the liver. Comparison of *P. vivax* prevalence during follow-up between the two study arms indicated that at least 50% of the *P. vivax* prevalence resulted from such relapses. Clinical episodes were more commonly caused by *P. falciparum* than by *P. vivax* despite a lower incidence of new clones in the blood. Increasing age was associated with reduced risk of clinical episodes with *P. vivax* and a concurrent or recent infection with *P. vivax* was associated with a reduced risk of a clinical episode with *P. falciparum*.

Essential revisions:

All the comments made in the reviews listed below should be addressed. In addition, following the reviewer consultation session, it was felt that authors should try to make the key advance of this work clearer. In particular, there should be a clear agenda articulated in the Introduction, including how this differs from the primary analysis which has already been published. Also, make clear in the Abstract the cohort you are analysing.

It was also noted that an a priori statistical analytical plan would have added a lot of credibility to the analysis and its conclusions, and the presence or absence of such a plan should be noted.

Reviewer #1:

This study followed a cohort of children in Papua New Guinea intensively to detect new infections by *Plasmodium falciparum* and *P. vivax* in the blood and to measure incidence of clinical disease caused by these parasites. These children had been initially cleared of blood stage malaria infection and sometimes also liver-stage *P. vivax*.

This is a rich dataset and a fascinating study and analysis which provides many important insights into the epidemiology of malaria and how the species interact. The manuscript is well written. The analysis and interpretation of such a complex dataset is generally excellent, though I do not agree with every aspect – see below for more detailed comments.

In particular, measuring the incidence of new infections by molecular methods is difficult because parasite densities often fluctuate below and above the detection limits of these methods (as previously shown by some of the same authors Koepfli, C., S. Schoepflin, M. Bretscher, E. Lin, B. Kiniboro, P. A. Zimmerman, P. Siba, T. A. Smith, I. Mueller and I. Felger (2011). "How much remains undetected? Probability of molecular detection of human Plasmodia in the field." PLoS One 6(17): e19010.) Thus the methods used by the authors could potentially overestimate the molecular force of infection, since infections which appear new are actually older ones with increased parasite density. This would influence the rest of the results. It is difficult to overcome this problem but it should be covered in more detail in the Discussion.

Reviewer #2:

The data are interesting and rich. The data set is unique – I am unaware of any other site with detailed molecular monitoring of Pv and Pf in a cohort studied in this way.

My major reservation with this particular analysis is that there isn't a clear advance articulated. The Abstract ends with results, and doesn't present a short description of what the most important conclusion is.

The Discussion is reasonably detailed but didn't guide me to a clear advance. The impact of hypnozoites on Pv risk has been described by previous work from the group on the impact of PQ in the randomized trial, the utility of _mol_FOB has been described in previous work, and spatial heterogeneity is described in detail elsewhere.

The link between asymptomatic acquisition of clones and febrile episodes is an important area to study, and showing links between the genotype of a febrile episode and the genotype of asymptomatic disease would have been interesting but these data are not presented.

Reviewer #3:

Hofmann and colleagues present a very interesting analysis from a cohort of children in PNG with in depth molecular analysis undertaken on blood samples collected every 14 days. The combination of dense sampling and 4 treatment arms with and without prolonged Pq radical cure, allows a unique opportunity to tease out the contributing factors of relapse and reinfection. The primary output of this study has already been published (Robinson et al. Plos Med 2015). In the current analysis the authors address the factors associated with greater risk of new infections and clinical episodes during follow up.

The findings once again highlight the high contribution of *P. vivax* to recurrent infections, and new insights into the epidemiology in these remote villages, the interaction (or lack thereof) between species. The paper is novel and interesting, and certainly a welcome addition to the literature. Although I have a number of comments:

Major Comments

1) The Robinson et al. paper is an excellent presentation of the primary analysis of this cohort containing many of the parameters included here. It would be good state clearly how the current analysis differs from this and the additional information it provides. The Abstract also ought to include the dates of the original study to make it clear to those reading the Abstract alone that this is the same core data as that presented previously.

2) The analysis is complex including multiple factors such as different diagnoses (LM and qPCR), clinical and asymptomatic infection, 4 species of infection, two schizontocidal treatments (AL and Cq), two radical cure arms (Pq and placebo), time to event (survival/Cox Regression) and incidence (generalized estimating equations). Multivariable analysis are presented, but the univariable analyses behind these models are mostly omitted. I believe the analysis was well conducted and appropriate but as a reviewer I was challenged to see how these multiple multivariable models were constructed. For instance Table 3 and Table 5 are Cox regression models and yet HR are stated – surely these are Adjusted Hazards Ratios (AHR)? Gaps in the tables suggest that these components weren't included in the final model, but no information is given of univariable factors underlying this and how these changed after controlling for other factors. Since these tables include four species it could be hard to bring in all the data, but supplementary tables to highlight all the factors would be welcome.

3) The multivariable models are also confounded by likely interactions in the terms. For instance, in Table 3 the Infection status at enrolment was correlated with the Village. Was an interactive term needed to address this?

4) In subsection “Risk of re-infection and molFOB during follow-up”. The term "reinfection" was confusing. In the Pq treatment arms, if one assumes complete clearance of the hypnozoite stages, then any recurrent parasitaemia might be expected to be reinfection. But otherwise recurrent parasitaemias could also be P.vivax relapses. It's not clear whether the analysis is just selecting the Pq arms and these are really reinfections or whether the authors should be calling this section Risk of Recurrent infection (which I believe is the correct term).

5) In subsection “Risk of re-infection and molFOB during follow-up”. The haemoglobin concentration and variation with sustained and recurrent infection is very interesting (and adds another layer of complexity). But it appears that the parameter stated in the final sentence and again in Table 4, is the Hb on enrolment. Or is this at other times during the follow up. A full analysis of the Hb changes over time (if taken) would be fascinating but I presume that may be another paper to come.

6) Discussion paragraph five. The authors suggest that Pv infection is greater after Pf. Interesting Table 4 suggest that new Pf increases the risk of Pv in the PQ arm (which should have no hypnozoites left) but not in the placebo arm (or implied by the gap in the table). But in Table 3, initial Pf had no effect on risk of Pv. The latter were asymptomatic infections and it is the fever associated with the Pf parasitaemia rather than the parasitaemia itself that is hypothesised to awaken the hypnzoites. Did the authors look at Pv FOB after febrile clinical episodes and whether this differed between the Pq and non Pq arms?

[Editors' note: further revisions were requested prior to acceptance, as described below.]

Thank you for resubmitting your work entitled "The complex relationship of exposure to new *Plasmodium* infections and incidence of clinical malaria in Papua New Guinea" for further consideration at *eLife*. Your revised article has been favorably evaluated by Prabhat Jha (Senior editor) and two reviewers, one of whom is a member of our Board of Reviewing Editors.

The manuscript has been improved but there are some remaining issues that need to be addressed before acceptance, as outlined below:

1) Time-varying exposures should be accounted for with time-varying covariates in the Cox model (as pointed out in the reviews of the original submission). This problem has not been adequately addressed.

2) Following consultation, the reviewers think the map (which is, apparently, for "illustrative purposes" rather than providing meaningful numerical estimates) should be moved to supplementary material (as a supplement to a primary figure, per journal policy) as this is not an essential part of the paper. The Kriging approach seems preferable to IDW for this.

3) *eLife* reporting guidelines state "Report exact p-values wherever possible alongside the summary statistics and 95% confidence intervals. These should be reported for all key questions and not only when the p-value is less than 0.05. " There are several places where this has not currently been done. Also, the current reporting and interpretation of p-values (dichotomising into significant/non-significant based on a.05 threshold) is inappropriate and inconsistent the recent ASA statement (http://dx.doi.org/10.1080/00031305.2016.1154108) which calls for a more nuanced interpretation.

4) There seems to be no justification for emphasising models based on stepwise variable selection. Such stepwise procedures are known to cause distortion and in this case appear unnecessary. The full models (currently reported in the supplementary material) should be reported instead.

Reviewer #2:

The authors have brought out a clear focus in the Abstract and Discussion, dealing with my previous major concern, and have in addition dealt with all my minor concerns extremely thoroughly.

The data are unique, the analysis is clear and insightful, and I would support publication.

On the minor point of kriging vs distance weighted interpolation the authors have presented a convincing case that it doesn't matter and I'm happy for whichever plot they feel most appropriate to be used – but it would be useful to include the other plot as supplementary material.

Reviewer #4:

1) Reviewer 2 raised the point that time varying exposures should be modeled using time varying covariates in Cox models. This is clearly correct, and biases that arise when this is not done have been well documented.

The authors' response to this was that "this was done because that variable can be estimated more reliably and shows less variation. In addition, there is also less risk of an auto-correlation with the outcome variable."

I don't think this response is adequate for the following reasons.

First, it is not clear what is meant by "auto-correlation with the outcome variable".

Auto correlation is correlation of a variable with lagged version of itself as the name suggests.

Second, it is not clear what they mean by "shows less variation" or why this should be important. Less variation between individuals? Why should that be a consideration?

Third, and most importantly, by including a covariate representing the mean _mol_FOB over the entire follow-up period the model is effectively saying that the risk of clinical malaria at any particular point of time depends on events that occur in the future. This is not consistent with what we know about how time works and is, in my opinion, indefensible. These problems can all be resolved by using a time varying covariates as reviewer 2 suggests. This is not difficult.

2) Reviewer 2 made another point about maps and the fact that the inverse distance weighted interpolation (IDW) method used by ArcGIS is not an appropriate method. This looks right to me, and the authors essentially agreed to this in their rebuttal letter but said: i) "It had not been our intention to conduct spatial risk mapping and therefore we had not used a model- based approach such as Kriging"; ii) a Kriging approached has now been used and results are similar to the IDW method; iii) "Considering the similarity between our original figure and the estimates obtained by the Kriging approach suggested by reviewer 2, we feel that IDW approach is sufficient to make this illustrative point and have thus kept the original figure in the revised manuscript file.

They added that "It should be noted that this is not a modelling method and the contours on the maps are for illustrative purposes rather than prediction or extrapolation."

My view on this is that contours are likely to misinterpreted as extrapolations (people will look at the pictures and forget the caveats), and if the results cannot be justified they shouldn't be in the paper. I am not clear what "for illustrative purposes" means. What is it illustrating? In my view the authors should just either show the raw spatial data, or present results of a model they can justify.

3) The biggest frustration for me with this paper was that it seems to be out of touch with mainstream statistical thinking in two key respects:

i) The authors repeatedly use the word "significant", without qualification, to dichotomise the interpretation of results depending on whether p-values are <0.05 or otherwise. This can only really be justified by the fact that it is common practice (which is not actually a justification at all), and it is certainly not in line with mainstream statistical thinking (see, for example, Ch 9 of Kirkwood & Sterne's Essential Medical Statistics 2nd Edition and the ASA's recent consensus statement on p-values http://amstat.tandfonline.com/doi/abs/10.1080/00031305.2016.1154108). Since arbitrarily small differences of no clinical significance will give arbitrarily small p-values with enough data, it is clearly more important to focus on effect sizes and associated CIs. However, in the current manuscript reporting is inconsistent: sometimes only point estimates and p-values are given (eg. Paragraph two of subsection “Patterns in the risk of *P. vivax* and *P. falciparum* clinical episodes”) while elsewhere CIs and p-values are reported. In my opinion it is essential to report CIs associated with point estimates, and p-values may be reported optionally.

ii) The use of stepwise variable selection procedures is known to cause problems (p-values will be too small, effect sizes too large – see, for example, Chapter 29 of Kirkwood & Sterne). It is also near universal advice to include covariates in a model where there are good a priori reasons for thinking they are likely to be important, regardless of what the p-values are. When you have enough data to include all these covariates there is no good reason for taking out variables because they fail to meet an arbitrary threshold of statistical significance (the fact that this is commonly done is not a good reason), and many good reasons not to. This problem can be easily fixed by reporting the full models (currently in the supplementary material) for Table 3, Table 4 and 5. I don't see any value in reporting the models with the automated variable selection, though the univariate models in the supplementary material are useful.

---

## [Author Response]

Essential revisions:All the comments made in the reviews listed below should be addressed. In addition, following the reviewer consultation session, it was felt that authors should try to make the key advance of this work clearer. In particular, there should be a clear agenda articulated in the Introduction, including how this differs from the primary analysis which has already been published. Also, make clear in the Abstract the cohort you are analysing.

We have rewritten the Abstract and key components of the Introduction and Discussion to ensure that these points are clear and that the key advance of this work is more clearly articulated.

It was also noted that an a priori statistical analytical plan would have added a lot of credibility to the analysis and its conclusions, and the presence or absence of such a plan should be noted.

An analytical plan focusses predominantly of the primary objectives of the trial – i.e. estimating the contribution of hypnozoites to the burden of infection, diseases and transmission – and has been included into the original trial protocol. This protocol has been registered on clinicaltrials.gov (NCT02143934) and is available upon request.

The analyses presented in this manuscript represents a more complex epidemiological analysis of secondary aims and additional endpoints of this study that were not covered in the initial analyses plan.

Reviewer #1:This study followed a cohort of children in Papua New Guinea intensively to detect new infections by Plasmodium falciparum and P. vivax in the blood and to measure incidence of clinical disease caused by these parasites. These children had been initially cleared of blood stage malaria infection and sometimes also liver-stage P. vivax.This is a rich dataset and a fascinating study and analysis which provides many important insights into the epidemiology of malaria and how the species interact. The manuscript is well written. The analysis and interpretation of such a complex dataset is generally excellent, though I do not agree with every aspect – see below for more detailed comments.In particular, measuring the incidence of new infections by molecular methods is difficult because parasite densities often fluctuate below and above the detection limits of these methods (as previously shown by some of the same authors Koepfli, C., S. Schoepflin, M. Bretscher, E. Lin, B. Kiniboro, P. A. Zimmerman, P. Siba, T. A. Smith, I. Mueller and I. Felger (2011). "How much remains undetected? Probability of molecular detection of human Plasmodia in the field." PLoS One 6(4): e19010.) Thus the methods used by the authors could potentially overestimate the molecular force of infection, since infections which appear new are actually older ones with increased parasite density. This would influence the rest of the results. It is difficult to overcome this problem but it should be covered in more detail in the Discussion.

Imperfect detectability of individual genotypes complicates measuring the incidence of new infections and indeed makes it difficult to exactly estimate the number of genetically distinct infections an individual acquires. As a consequence – and as outlined in both Mueller et al. PNAS, 2012 and Koepfli et al. PLoS NTD, 2013, _mol_FOB is a proxy rather than a direct measure of exposure. Whether _mol_FOB over- or under-estimates the true number of newly acquired infections is dependent not only on the detectability of individual clones, but also on the mean duration (relative to sampling frequency) of an individual blood-stage infection. Generally, short infection durations will lead to an underestimation of the true incidence of infection, i.e. many infections will not be detected at all. Long durations may result in an over-estimation of the true incidence of infection, although we partially account for this by classifying a re-occurrence of the same clone as a new infection only when not detected in two preceding samples, opposed to a continuing infection (undetected for <2 immediately preceding samples).

Importantly for the results presented in our current manuscript, _mol_FOB (although a proxy measure of the true number of infections acquired by an individual) is relatively unbiased in regards to the most important variables influencing the risk of acquiring new infections such a season, age or the use of malaria control interventions. _mol_FOB is therefore an unbiased estimate of difference in exposure between different individuals in the same cohort. Hence, even if we cannot exactly quantify the difference between the true incidence of new infection and its proxy measure, _mol_FOB, this difference will not affect our ability to correctly estimate the effects of other variables on exposure and/or transmission.

We have now included a paragraph in the Discussion addressing the issue of imperfect detectability and its relevance for the results presented here.

Reviewer #2:The data are interesting and rich. The data set is unique – I am unaware of any other site with detailed molecular monitoring of Pv and Pf in a cohort studied in this way.My major reservation with this particular analysis is that there isn't a clear advance articulated. The Abstract ends with results, and doesn't present a short description of what the most important conclusion is.The Discussion is reasonably detailed but didn't guide me to a clear advance. The impact of hypnozoites on Pv risk has been described by previous work from the group on the impact of PQ in the randomized trial, the utility of _mol_FOB has been described in previous work, and spatial heterogeneity is described in detail elsewhere.

According to the panel of reviewer’s recommendations we have now re-written the Abstract and parts of the Discussion to emphasize the novelty of the present manuscript, which consists in – for the first time – bringing together all of the above three subjects in one comprehensive, in-depth analysis. In particular, this is the first time that _mol_FOB for *P. falciparum* and *P. vivax* has been assessed and compared in a cohort of children where we were able to selectively remove the contribution of *P. vivax* relapsing infections from half of the children. The fact that we can directly compare patterns of *P. falciparum* and *P. vivax* new mosquito-derived infections versus the patterns observed from *P. vivax* relapse represents a clear advance upon previous work.

For better guidance of reader, we have added a paragraph to the Introduction outlining how this manuscript advances on the results from each preceding paper. We revisit this structure in the Discussion section, where we refer to each of the key papers published previously and reflect on how our detailed molecular genotyping on samples combined with a novel treatment study design contributes to an in-depth understanding of malaria epidemiology in PNG and in general.

The link between asymptomatic acquisition of clones and febrile episodes is an important area to study, and showing links between the genotype of a febrile episode and the genotype of asymptomatic disease would have been interesting but these data are not presented.

This was also pointed out by reviewer 1. As stated above, we agree with the reviewers that it would be most interesting to investigate in more detail the links between genotypes of febrile episodes versus asymptomatic infections in this dataset. We have now included data on the proportion of episodes that carry new infections in the Results section. An in-depth analysis of clonal patterns in clinical versus purely asymptomatic infections requires additional parasite genotyping and presents a highly complex analysis. We therefore feel that is beyond the scope of a revision for this manuscript.

Reviewer #3:[…]Major Comments1) The Robinson et al. paper is an excellent presentation of the primary analysis of this cohort containing many of the parameters included here. It would be good state clearly how the current analysis differs from this and the additional information it provides. The Abstract also ought to include the dates of the original study to make it clear to those reading the Abstract alone that this is the same core data as that presented previously.

This point has been jointly raised by all reviewers, and we have rewritten the Abstract, as well as parts of the Introduction and Discussion in order to clarify how this manuscript advances over the Robinson et al. paper as well as other previous papers investigating _mol_FOB in PNG children.

2) The analysis is complex including multiple factors such as different diagnoses (LM and qPCR), clinical and asymptomatic infection, 4 species of infection, two schizontocidal treatments (AL and Cq), two radical cure arms (Pq and placebo), time to event (survival/Cox Regression) and incidence (generalized estimating equations). Multivariable analysis are presented, but the univariable analyses behind these models are mostly omitted. I believe the analysis was well conducted and appropriate but as a reviewer I was challenged to see how these multiple multivariable models were constructed. For instance Table 3 and Table 5 are Cox regression models and yet HR are stated – surely these are Adjusted Hazards Ratios (AHR)? Gaps in the tables suggest that these components weren't included in the final model, but no information is given of univariable factors underlying this and how these changed after controlling for other factors. Since these tables include four species it could be hard to bring in all the data, but supplementary tables to highlight all the factors would be welcome.

All model estimates are given as adjusted hazard ratios (AHR), which we have now clarified in the tables.

As stated in the Materials and methods section, for each analysis the most parsimonious model was generated from a full model by back-selection. We have now included supplementary tables with all full models. These complement the back-selected models presented in the main paper to allow a better evaluation of model results.

Given the basic design of the trial and the strong effect of the randomized PQ treatment on the *P. vivax* burden, we feel it would be inappropriate to present results of *P. vivax* ‘univariate’ analyses that are not adjusted for the PQ treatment effect. Although the treatment does not affect the *P. falciparum* burden in the same way, we also adjusted analyses of potential risk factors for *P. falciparum* infection and disease for PQ treatment in ‘univariate’ analyses. These treatment-only adjusted HRs, rather than purely univariate factors, for each of the variables are now presented in an additional supplementary table for each main table.

3) The multivariable models are also confounded by likely interactions in the terms. For instance, in Table 3 the Infection status at enrolment was correlated with the Village. Was an interactive term needed to address this?

No. We tested extensively for interactions and are confident that we have explored potential confounding factors in sufficient detail in our preliminary analyses. Given the large amount of data in the manuscript, we only present the final, most parsimonious models and their results. We have now added a sentence to the statistical methods section highlighting that we did test for interactions in preliminary analyses and no significant interactions were found.

4) In subsection “Risk of re-infection and molFOB during follow-up”. The term "reinfection" was confusing. In the Pq treatment arms, if one assumes complete clearance of the hypnozoite stages, then any recurrent parasitaemia might be expected to be reinfection. But otherwise recurrent parasitaemias could also be P.vivax relapses. It's not clear whether the analysis is just selecting the Pq arms and these are really reinfections or whether the authors should be calling this section Risk of Recurrent infection (which I believe is the correct term).We agree with the reviewer and have updated this throughout the manuscript, now referring to recurrent infections or risk of recurrent infection.5) In subsection “Risk of re-infection and molFOB during follow-up”. The haemoglobin concentration and variation with sustained and recurrent infection is very interesting (and adds another layer of complexity). But it appears that the parameter stated in the final sentence and again in Table 4, is the Hb on enrolment. Or is this at other times during the follow up. A full analysis of the Hb changes over time (if taken) would be fascinating but I presume that may be another paper to come.

Table 4 investigates parameters associated with the incidence of new clones per follow-up sampling interval, i.e. the detection of a new clone since the preceding sampling visit. In this model, haemoglobin concentration is included as the haemoglobin concentration at the end of the sampling interval; i.e. our data show that a reduced heamoglobin level at a next sampling visit is associated with a newly acquired *P. falciparum* infections in the preceding interval. This is suggestive of an in highly exposed individuals; however, no longitudinal analysis of changing haemoglobin levels during follow-up was performed. We are indeed planning much more detailed analyses of Hb changes over time and their relationship with *Plasmodium* infections using data from this and some of our other cohorts. Results from these analyses will be presented elsewhere.

6) Discussion paragraph five. The authors suggest that Pv infection is greater after Pf. Interesting Table 4 suggest that new Pf increases the risk of Pv in the PQ arm (which should have no hypnozoites left) but not in the placebo arm (or implied by the gap in the table). But in Table 3, initial Pf had no effect on risk of Pv. The latter were asymptomatic infections and it is the fever associated with the Pf parasitaemia rather than the parasitaemia itself that is hypothesised to awaken the hypnzoites. Did the authors look at Pv FOB after febrile clinical episodes and whether this differed between the Pq and non Pq arms?

As stated above, Table 4 investigates parameters associated with the incidence of new clones during follow-up per follow-up interval, while Table 3 investigates predictors of the time-to-reinfection with *P. vivax*. In both tables, infections are included irrespective of whether they were symptomatic or asymptomatic at time of sampling.

From Table 4 we conclude that a new infection with *P. falciparum* is associated with a new infection with *P. vivax* in the same follow-up intervalin the PQ arm, where the vast majority of new *P. vivax* infections originate from mosquito bites. Here, both species are potentially transmitted within one single infective bite or by several mosquito bites in heavily exposed children. This association is absent in the placebo arm, where most *P. vivax* infections originate from relapsing hypnozoites and only proportionally few new blood-stage infections are associated with an infective bite.

We did not specifically investigate *P. vivax*_mol_FOB after febrile episodes in this analysis; however, a novel mathematical modeling approach applied to data from this cohort and data from other *P. vivax*-endemic countries has shown that there is no observable association between relapses and fevers (White et al., manuscript under review, PLoS Comp Biol). We have now included a reference in the Discussion that more in-depth modelling will be required to determine the association of *P. vivax* replaces with different proposed relapse triggers.

[Editors' note: further revisions were requested prior to acceptance, as described below.]

The manuscript has been improved but there are some remaining issues that need to be addressed before acceptance, as outlined below:1) Time-varying exposures should be accounted for with time-varying covariates in the Cox model (as pointed out in the reviews of the original submission). This problem has not been adequately addressed.

To address the reservations of the reviewers, we have now adjusted the way we include _mol_FOB as a covariate in Cox models (Table 5). Instead of calculating _mol_FOB over the whole follow-up period, resulting in a stationary measure of exposure for each child, we now calculate _mol_FOB up to each follow-up visit (i.e. the average _mol_FOB estimate as 'known' at the time of 'outcome'). Using this approach, our estimate of _mol_FOB is time-dependent and the risk of clinical episodes is predicted using only exposure up to the time point of observation, without taking into account future exposure.

This approach is described in the Materials and methods section: “In the GEE and Cox models where _mol_FOB was a covariate, it was included as a time-varying covariate. When modelling _mol_FOB (Table 4) and the odds of clinical episodes (Table 6) using GEEs, _mol_FOB was calculated for each follow-up interval and used as predictor. In Cox models investigating the risk of clinical episodes (Table 5), _mol_FOB was calculated based on the new infections up to the time of failure and used as predictor.”

2) Following consultation, the reviewers think the map (which is, apparently, for "illustrative purposes" rather than providing meaningful numerical estimates) should be moved to supplementary material (as a supplement to a primary figure, per journal policy) as this is not an essential part of the paper. The Kriging approach seems preferable to IDW for this.

We have produced new maps using the kriging approach as preferred by the panel of reviewers. Details of the kriging parameters are specified in the Materials and methods section: “Maps were drawn using Arcgis 10.1 (Esri Inc.). Ordinary kriging was used to generate the contour maps. Semivariograms were used as the mathematical forms used to express autocorrelation. Input variables for the models were i) _mol_FOB as measured per individual (if multiple individuals per household existed, the mean was used), resulting in Figure 3 Panels A and B; ii) relative risk of clinical episodes as predicted by the model shown in Table 5, resulting in Panels C and D of Figure 3. Relative standard error maps were generated by dividing the absolute standard error map by the model prediction map.”

A main focus of the manuscript consists in describing small ‐scale geographical heterogeneity in exposure to different Plasmodium species, and investigating how these differences in exposure translate to patterns in clinical incidence. To illustrate the geographical differences in exposure and clinical incidence between villages, we would prefer to keep the kriging maps in the main manuscript.

However, if substantial concerns about presenting kriging ‐fitted maps in the main manuscript (along with the relative standard error) remain within the panel of reviewers, these maps may be moved to the supplement in the final version of the manuscript.

3) eLife reporting guidelines state "Report exact p-values wherever possible alongside the summary statistics and 95% confidence intervals. These should be reported for all key questions and not only when the p-value is less than 0.05. " There are several places where this has not currently been done. Also, the current reporting and interpretation of p-values (dichotomising into significant/non-significant based on a.05 threshold) is inappropriate and inconsistent the recent ASA statement (http://dx.doi.org/10.1080/00031305.2016.1154108) which calls for a more nuanced interpretation.

In addition to including the full models in the main text to avoid selective reporting based on a cutoff of p<0.05, we have reformulated parts of the Results section giving more emphasis to effect sizes rather than p-values. We now give all confidence intervals also in the text with every point estimate to facilitate interpretation by the reader.

4) There seems to be no justification for emphasising models based on stepwise variable selection. Such stepwise procedures are known to cause distortion and in this case appear unnecessary. The full models (currently reported in the supplementary material) should be reported instead.

We have now moved the full models from the supplement to the main text, and provide univariate (or in selected cases PQ treatment-adjusted factors) in the supplement.

Reviewer #2:The authors have brought out a clear focus in the Abstract and Discussion, dealing with my previous major concern, and have in addition dealt with all my minor concerns extremely thoroughly.The data are unique, the analysis is clear and insightful, and I would support publication.

We thank the reviewer for their assistance in improving the manuscript and we are pleased that we have adequately dealt with all concerns.

On the minor point of kriging vs distance weighted interpolation the authors have presented a convincing case that it doesn't matter and I'm happy for whichever plot they feel most appropriate to be used – but it would be useful to include the other plot as supplementary material.

In response to the Senior Editor and reviewer 4 we have now included the kriging maps in the main manuscript file, although we remain willing to move to the supplementary material if necessary.

Reviewer #4:1) Reviewer 2 raised the point that time varying exposures should be modeled using time varying covariates in Cox models. This is clearly correct, and biases that arise when this is not done have been well documented.The authors' response to this was that "this was done because that variable can be estimated more reliably and shows less variation. In addition, there is also less risk of an auto-correlation with the outcome variable."I don't think this response is adequate for the following reasons.First, it is not clear what is meant by "auto-correlation with the outcome variable".Auto correlation is correlation of a variable with lagged version of itself as the name suggests.Second, it is not clear what they mean by "shows less variation" or why this should be important. Less variation between individuals? Why should that be a consideration?Third, and most importantly, by including a covariate representing the mean molFOB over the entire follow-up period the model is effectively saying that the risk of clinical malaria at any particular point of time depends on events that occur in the future. This is not consistent with what we know about how time works and is, in my opinion, indefensible. These problems can all be resolved by using a time varying covariates as reviewer 2 suggests. This is not difficult.

We have now included _mol_FOB in the models as time-varying covariate, as outlined above.

2) Reviewer 2 made another point about maps and the fact that the inverse distance weighted interpolation (IDW) method used by ArcGIS is not an appropriate method. This looks right to me, and the authors essentially agreed to this in their rebuttal letter but said: i) "It had not been our intention to conduct spatial risk mapping and therefore we had not used a model- based approach such as Kriging"; ii) a Kriging approached has now been used and results are similar to the IDW method; iii) "Considering the similarity between our original figure and the estimates obtained by the Kriging approach suggested by reviewer 2, we feel that IDW approach is sufficient to make this illustrative point and have thus kept the original figure in the revised manuscript file.They added that "It should be noted that this is not a modelling method and the contours on the maps are for illustrative purposes rather than prediction or extrapolation."My view on this is that contours are likely to misinterpreted as extrapolations (people will look at the pictures and forget the caveats), and if the results cannot be justified they shouldn't be in the paper. I am not clear what "for illustrative purposes" means. What is it illustrating? In my view the authors should just either show the raw spatial data, or present results of a model they can justify.

We have now included the kriging maps, as outlined above.

3) The biggest frustration for me with this paper was that it seems to be out of touch with mainstream statistical thinking in two key respects:i) The authors repeatedly use the word "significant", without qualification, to dichotomise the interpretation of results depending on whether p-values are <0.05 or otherwise. This can only really be justified by the fact that it is common practice (which is not actually a justification at all), and it is certainly not in line with mainstream statistical thinking (see, for example, Ch 9 of Kirkwood & Sterne's Essential Medical Statistics 2nd Edition and the ASA's recent consensus statement on p-values http://amstat.tandfonline.com/doi/abs/10.1080/00031305.2016.1154108). Since arbitrarily small differences of no clinical significance will give arbitrarily small p-values with enough data, it is clearly more important to focus on effect sizes and associated CIs. However, in the current manuscript reporting is inconsistent: sometimes only point estimates and p-values are given (eg. Paragraph two of subsection “Patterns in the risk of P. vivax and P. falciparum clinical episodes”) while elsewhere CIs and p-values are reported. In my opinion it is essential to report CIs associated with point estimates, and p-values may be reported optionally.

We have now rewritten the Results section to give more emphasis to effect sizes rather than p-values and we provide all confidence intervals in the text with every point estimate to facilitate interpretation by the reader.

ii) The use of stepwise variable selection procedures is known to cause problems (p-values will be too small, effect sizes too large – see, for example, Chapter 29 of Kirkwood & Sterne). It is also near universal advice to include covariates in a model where there are good a priori reasons for thinking they are likely to be important, regardless of what the p-values are. When you have enough data to include all these covariates there is no good reason for taking out variables because they fail to meet an arbitrary threshold of statistical significance (the fact that this is commonly done is not a good reason), and many good reasons not to. This problem can be easily fixed by reporting the full models (currently in the supplementary material) for Table 3, Table 4 and 5. I don't see any value in reporting the models with the automated variable selection, though the univariate models in the supplementary material are useful.

As outlined in response to Senior Editor comment 4, we have now moved the full models from the supplement to the main text, and provide univariate (or in selected cases PQ treatment-adjusted factors) in the supplement.